



Title page

## 2 **Back to the future- Conservative grassland management for**

## 3 **Anthropocene soils in the changed landscapes of Uruguay?**

Ina Säumel[1], Leonardo R. Ramírez[1], Sarah Tietjen[2], Marcos Barra[3] and Erick Zagal[4]
[1] Integrative Research Institute THESys Transformation of Human-Environment-Systems
Humboldt-Universität zu Berlin, Unter den Linden 6, Berlin, 10099, Germany.
[2] Leibniz Institute of Vegetable and Ornamental Crops (IGZ) e.V., Theodor-Echtermeyer-Weg
1, 14979 Großbeeren, Germany
[3] Städtisches Klinikum Dessau, Auenweg 38, 06847 Dessau-Roßlau, Germany
[4] Departamento de Suelos y Recursos Naturales, Universidad de Concepción, Campus Chillán,
Vicente Méndez 595, Chillán, Chile
*Correspondence to*: Ina Säumel (ina.saeumel@hu-berlin.de)
**Abstract.** The 'soils of the anthropocene' are predominately agricultural. To understand them,
we analysed agri- and silvicultural intensification of Uruguayan grasslands in a country wide
survey on fertility proxies, pH and trace metals in topsoils originating from different land uses.
We observed a loss of nutrients, trace metals and organic matter from grassland, crops and
timber plantations, and its accumulation in the topsoils of riverine forests. The translocation of
nutrients and organic matter across the landscape to the erosion base depends on local land use
trajectories. Increasing soil acidification is driving a positive feedback loop, and land use
intensification is leading to degradation of local black soils within a few decades. Our data
raises questions about the resilience and carrying capacity of Uruguayan soils with regard to
currently implemented highly productive management forms, including the use of timber



plantation for carbon sequestration, and supports more conservative forms of extensive
management on the grassland biome.

# 1. Introduction

Human activities alter the bio- and pedosphere, leaving a footprint of such a magnitude that it
can be verified stratigraphically (Waters et al., 2016). This unprecedented transformational
force is intimately related to the expansion of societies and its productive frontiers, causing a
loss of biodiversity, habitat and soil degradation and, consequently, to ecosystem modification
(Foley et al., 2005, Borrelli et al., 2017). In this context, soil sciences have transitioned from
studies on natural soil formation to the science of 'anthropedogenesis' (Richter, 2020),
focussing on the 'soils of the anthropocene' that are predominately agricultural (51 Mio. $km^2$)
or urban (1.5 Mio $km^2$; FAO, 2019).
The temperate grasslands of South America have historically been characterised by rolling
plains and low hills that have been extensively exploited for cattle production and its
derivatives since the arrival of European colonization. The Río de la Plata grasslands are one
of our planet's four major black soil regions (Durán et al., 2011; Liu et al., 2012), and some of
the most fertile soils in the world. Playing an important role in the global food production, these
are characterized by a thick, humus and base cation rich and a high cation exchange capacity
throughout their profile. Maintaining their properties are therefore crucial to developing
sustainable and productive agriculture (Durán et al., 2011; Liu et al., 2012).
Today, the 'Uruguayan savanna' is part of the top-three most critically endangered biomes
(Veldman et al., 2015). In recent decades, however, the grassland has decreased due to the
expansion of cash crops and *Eucalyptus* plantations, both of which are promoted by national
legislation including land grabbing and trans-nationalization (Piñeiro, 2012). This land use

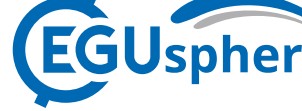

intensification, with its increased input of energy, nutrients and pesticides, leads to an overall
loss of soil fertility and increasing toxicity related to acidification, salinization and
contaminants (Liu et al., 2012; Borrelli et al., 2017). Ecological, economic and cultural
functions of soils are severely degraded, and the degradation of black soils in South America
is of particular concern because they have only been heavily exploited for a comparatively
short period of time (Durán et al., 2011).
Since the first decades of the twentieth-century, compared to other disciplines, soil sciences
have received an extraordinary amount of attention in Uruguayan academia, governance, the
productive sector, and also in the general public, resulting in a national soil inventory program
in 1965. The subsequent classification of Uruguayan soils and their productivity (CONEAT
Index) remains an important source for today's land taxation and for management plans by the
legal conservation regulations, and provides a detailed classification that takes into account soil
type, texture, natural vegetation, altitude and geology (Lanfranco and Sapriza, 2011).
As soil degradation is extremely relevant for countries like Uruguay, which are
socioeconomically dependent on their soils (Zubriggen et al. 2020), it is a topic of discussion
for local farmers, academia, and the public. An actualization of the state of the art of soils and
related processes is needed (García-Préchac et al. 2004; De Faccio et al. 2021), particularly as
there has been little study of the impacts of the Uruguayan grassland intensification on soils
properties (Beretta-Blanco et al., 2019). At the same time, while a new paradigm for grassland
intensification with a wide set of means including fertilization has been proposed to increase
economic and environmental sustainability (Jaurena et al., 2021), it is urgent to get more
insights into the dynamics of nutrient in soils of Uruguay and their availability for crops
(Beretta-Blanco et al., 2019).
Soil classifications are mainly based on subsoils. However, we focus on topsoil as the most
relevant and very responsive interface for ecological processes and farmers management, since
understanding the state of the art of topsoils and its processes is crucial for developing



recommendations for sustainable land management practices. Due to the diversity of
perspectives on soil quality and health and related ecosystem services, operational procedures
for evaluation of soil functioning are still lacking (Bonfante et al. 2021). We contribute to a
better understanding of globally occurring degradation processes in the field of tension between
desired soil productivity, yield limits, especially in erosion sensitive soils, and necessary soil
conservation.
We therefore explored soil parameters describing actual chemical conditions of topsoils that
are parts of different soil groups and orders, and Uruguayan soil categories. Specifically, in
order to explore the gains and losses of macro and micro-nutrients and soil organic matter
across landscapes and to determine the impact of land use change on acidification and trace
metal mobility and related trade-offs with soil degradation and conservation, we analysed i)
the variation of fertility proxies, ii) levels of acidification (pH), and iii) trace metals
accumulation.

## 86  2. Material and Methods

### 87  2.1 Study area and design

Uruguay covers about 176,000 km$^2$, and has a population of 3.5 million, mainly in urban areas.
The Western half of Uruguay is dominated by Mollisols, developed on a wide range of
sediments from the Devonian to the Cenozoic era (partially associated with Vertisols), while
the Eastern plains and wetlands have, in addition to Mollisols, other soil types on the slopes,
rocks and on flood plans (i.e. Alfisols, Ultisols, Entisols and Inceptisols; Durán et al. 2011).
During a country-wide survey from December 2015 to March 2016, we collected 280 topsoil
samples of 0-10 cm depth from 101 plots (50x50m in grassland and crops; 100x100m in native
forests and timber plantations) distributed at 28 monitoring sites throughout Uruguay, South



America (Fig. 1a-b), using a stratified random design. In the first step, we randomly selected
monitoring sites across the country. In the second step, we contacted landowners to explore
their willingness to establish a long-term monitoring site. If the owner agreed, plot selection
was stratified by different rural land use types: grassland, timber plantation, native forest, and
crops. We sampled top soil three times at each land use at the edges of the plot, and stored
samples below 7°C until lab processing.

## 2.2 Analysis of Soil Samples

For gravimetric determination of humidity, the topsoils samples were dried at 105 °C until
constant weight. Next, lumps in the samples were broken down and the remaining plant
material was removed before sieving (2 mm) and ground.
We analysed 280 samples regarding macronutrients, pH and trace metals and 80 samples for
soluble cations and micronutrients. Among fertility-related variables, we measured the total
amount of the macronutrients phosphorus (P), carbon (C) and nitrogen (N), so obtaining the
C/N ratio. To determine total carbon and nitrogen, the samples were sieved again (0.5 mm) and
analyzed using a LECO TruSpec CN (USA) at a combustion temperature of 950 °C with ultra-
pure oxygen. In addition, the presence of inorganic carbon was tested for by adding
concentrated hydrochloric acid, the presence of carbonates was assessed to obtain the amount
of organic carbon ($C_{org}$) and soil organic matter (SOM) as $C_{org}$ x 2 (Chenu et al. 2015).
We determined concentrations of the soluble cations calcium (Ca), magnesium (Mg),
potassium (K), sodium (Na), and the micronutrients copper (Cu), zinc (Zn), manganese (Mn)
and iron (Fe) by atomic spectroscopy (Unicam AAS Solaar 969, Thermo Electron Corporation,
USA), extracting with ammonium acetate (1 mol l$^{-1}$, pH 7), and with DTPA-CaCl$_2$-TEA at (pH
7.3). We calculated the cation-exchange capacity (CEC). Acidity was measured by adding
calcium chloride (0.01 mol) to the samples in a 2.5:1 proportion, and after shaking and two



hours rest, read with a pH meter (HI2550 meter, Hanna Instruments, USA)). For these
variables, each tenth sample was duplicated. We categorized the values using the USDA
Natural Resources Conservation Service classification (Kellogg, 1993).
To determine the concentrations of arsenic (As), cadmium (Cd), chromium (Cr) and lead (Pb),
samples were further sieved as before (0.5 mm), weighed out into a digesting container, and
extracted with a mixture of nitric acid (70%) and hydrochloric acid (30%) in a 1:3 proportion.
The digestion took place in a Titan MPS (Perkin Elmer, USA) microwave at a programme
suitable for this digestion (Method 3051A). During the proceedings, only deionized water Type
I ASTM1193 (EC max 0.06 to 0.1 µS/cm) was used. Reagents were used to eliminate traces of
other materials and to avoid contamination of the samples. The trace metals were determined
by inductively coupled plasma optical emission spectroscopy using the Optima 8000 (Perkin
Elmer) with metal-associated wavelengths of 193.696 nm for As, 228.802 nm for Cd, 267.716
nm for Cr and 220.353 nm for Pb.
Total P concentration of phosphorus was determined calorimetrically after microwave-assisted
digestion with Unicam spectrometer at a wavelength of 660 nm. For all these variables, a
repetition for each round of the microwave digestion was made.

## 2.3 Soil classification

We intersected the coordinates of the centre of the plots with maps containing geospatial
information on the classification of the Uruguayan soils using ArcGIS 10.3 (ESRI, 2018). For
Soil Groups classification, we used the of the World Reference Base for Soil Resources (WRB;
IUSS Working Group, 2015); for Soil Orders, the USDA soil taxonomy (Soil Survey Staff,
1999); and for the local Uruguayan classification, Soil CONEAT (Comisión Nacional de
Estudio Agronómico de la Tierra) Groups categories, which include productive capacity of
cattle and sheep (MGAP, 2021).



## 2.4 Data Analysis


In a first step, we explored and prepared our database for further analysis. Exploring the
distribution of the soil parameters in R (R Core Team 2021), ruled out the normality of the data
using the Shapiro-Wilk Test and the homogeneity of variances with the Fligner-Killeen Test.
We tested for outliers using the 1.5-3 IQR threshold and the function *outlierTest* from the R
package *car* (Fox and Weisberg 2019), reviewing the flagged observations case by case in the
experimental context. The variables on soils characteristics showed generally positive skewed
distributions, in some cases multimodal, and tests showed evidence contrary to assumptions of
spatial autocorrelation, homoscedasticity and normality in most cases (Supplementary
Material: Table S1-S2; Fig. S1-S2).
Spearman's rank correlations ($\rho$) were calculated to explore linear associations between soil
parameters across all single samples and within different land uses. We used the *adonis*
function of the R package *vegan v2.5-7* (Oksanen et al., 2020) with a Euclidean dissimilarity
matrix from our normalized soil data to perform PERMANOVA tests with 9999 permutations
to analyse the multivariate homogeneity of group dispersions based on differences on soil
parameters between land uses.
To compare the general effects of land use type on topsoil parameters, non-parametric Kruskal-
Wallis tests were carried out in R. When significant ($p \leq 0.05$), we used Pairwise Wilcoxon
Rank Sum tests with Benjamini & Hochberg correction to evaluate pairwise differences among
land uses.
We used non-metric multidimensional scaling (NMDS) as a robust unconstrained ordination
method to visualize patterns of top soil characteristics among all samples and within
subsamples (intersected with different soil Orders) across different land uses using the Bray-
Curtis dissimilarity matrix. The matrix was constructed with the *vegan* package to depict
patterns of all soil parameters in two dimensions (Fig.3a, c-f) and for the dataset without soluble



cations and micronutrients variables comparing subcategories within single land use types (i.e.
within grasslands in 'undisturbed', 'partially grazed' and 'highly grazed' plots; timber
plantations in '*Eucalyptus*' or '*Pinus*' plots; Fig. 3b).

# 3. Results

Nearly 75% of the sampled 101 plots intersected with the soil Group category 'Luvic Phaeozem'
(WRB; IUSS Working Group, 2015). A quarter of our plots were distributed among four other
Groups (i.e. Haplic Luvisols, Eutric Vertisols, Planosols, Lithic Leptisols; Fig. 1c). Half of our
plots intersected with the Orders 'Argiudolls' or 'Argiudolls & Hapluderts' (Soil Survey Staff,
1999), and about a quarter with 'Hapludalfs & Hapludults' or 'Argiudolls, Hapludolls &
Hapludalfs' (Fig. 1d). Our plots intersected with 32 different CONEAT categories (MGAP,
2021). The most frequent categories were '10.2', '9.3' and '2.12' (25% of our plots) and another
quarter with '9.1', '8.8', '9.6' and 'G03.11' (Fig. 1e).

## 3.1 General characteristics of Uruguayan topsoils

The measured topsoil parameters vary widely across Uruguay, between the different land uses
and classification to different soil orders (Table 1, Table A1-A4). The soil organic matter in
topsoils ranged between 0.8 to 16 percent, and was highest in native forests and lowest in timber
plantations. The mean of the C/N ratio was about 13, and lowest in crop soils. Phosphorous
ranged between 43 to 1009 mg kg$^{-1}$. We also observed a high variability for the micro- and
macronutrients and trace metals. The average cation-exchange capacity (CEC) was 12.41 cmol
kg$^{-1}$, highest in topsoils of native forest, followed by crops and grasslands and timber
plantations (Table 1, Table A1-A2).
For the whole data set, high correlation was found between P with SOM and Zn ($\rho=0.82$ and
0.76, respectively), and between Mg with Ca and Na ($\rho=0.82$ and 0.76, respectively; Fig. 2).



Similar results were observed within particular land uses, although in native forests, a negative
moderate correlation between Ca and Fe was observed (Fig. S3). There was high correlation
between pH and Ca ($\rho$=0.89; Fig. 2). In native forests, we also found similar correlation with
other soluble cations like Mg and K (Fig. S3). In park forests topsoils, there was a negative
correlation between pH and As and Pb ($\rho$=-0.81 and $\rho$=-0.84, respectively). In highly grazed
pastures and crops pH was highly correlated with Cr, and in crops also with As ($\rho$=0.93; Fig.
S3). Among trace metals, Spearman correlation was moderate between As with Cr and Pb and
high between Cr and Pb (Fig. 2). These correlation trends increased in grasslands, timber
plantations and crops. We also found a high correlation between cadmium and Cr in crops
($\rho$=0.81). Phosphorus was highly correlated with Cr and As in pine plantations, while in
soybean crops the correlation was between phosphorus with Cr and Pb (Fig. S3).
**3.2 Topsoil characteristics clustered by land use**
We found differences in multivariate distribution of samples according to the different land
uses (i.e., grassland, timber plantations or native forests; Fig. 3a-f; Table S3-S8). All pairwise
comparison of land uses showed significant differences with all variables (p=0.0001; Fig. 3a;
Table S3). We analysed subcategories within a land use type using the dataset without soluble
cations and micronutrients variables, only finding significant differences between *Eucalyptus*
and *Pinus* stands (*p*=0.0001; Fig. 3b; Table S4) but not among different grassland subtypes.
We also found differences analysing subsamples of different soil order classification to the
different land uses (Fig. 3c-f; Table S5-S8). The samples intersected with 'Argiudolls' included
all plots on crops, and we found significant differences in all pairwise comparisons of land uses
except between grasslands and timber plantation (*p*=0.0004; Fig. 3c; Table S5). We further
found differences between timber plantations and native forests at soils of the 'Argiudolls &
Hapluderts' Orders (*p*=0.0009; Fig. 3d; Table S6) and between timber plantations and
grasslands or native forests in soils of the 'Argiudolls, Hapudolls & Hapludalf' Orders



(*p*=0.0284; Fig. 3e; Table S7). Results for samples in 'Hapludalfs & Hapludults' soils were
similar to those obtained at country scale (*p*=0.0001; Fig. 3f; Table S8).

## 3.3 Differences in fertility proxies

We found significantly higher values for the fertility proxies for SOM, P, Ca, Mg and Zn in
topsoils from native forests compared to grasslands and timber plantations (Fig. 4b-e, h).
Phosphorous was significantly higher in topsoils of grassland compared to those of timber
plantations (Fig. 4c). Potassium was significantly higher in topsoils of native forests compared
to timber plantations (Fig. 4f). At subcategories level, we found significantly higher amounts
of K in partially used grasslands in comparison to samples from highly grazed pastures, and
higher values of SOM (*p*=0.002), P (*p*=0.059), Na (*p*=0.043), K (*p*=0.012) and Zn (*p*=0.048)
in *Eucalyptus* compared to *Pinus* plantations (Fig. S4). Among native forests, samples from
riverine forests contain more Mg (*p*=0.023) and Na (*p*=0.023) in comparison to park forests.
Considering all samples within the Order 'Argiudolls', C/N ratio in agricultural topsoils are
lower compared to all other land uses. Phosphorus was higher in topsoils of native forests and
crops compared to grasslands and timber plantations. Soil organic matter was highest in native
forests (Fig. 5a-o; Table A3-A4).

## 3.4 Soil Acidification

Topsoil samples showed a markedly acid profile (median pH=4.66), with nearly 75% classified
as 'extremely acidic' and 'very strongly acidic' (Fig. 6a). We found significant differences across
land uses, with less acidic values in native forests, and lower pH in timber plantations (Fig.
6b). Comparing between land uses, we found more samples with neutral acidity in grasslands
and more with higher acidity in timber plantations (Fig. 6b-c). In addition, we found lower pH
in samples from *Pinus* compared to *Eucalyptus* stands (p=0.018). Results of analysis inside soil
Orders showed similar variations observed at country scale, with timber plantations being more
acid and native forest closer to neutral pH (Fig. 6d-f).



## 3.5 Trace metal accumulation across land uses

For As, Cd, Cr and Pb, we found significantly higher values in topsoils originating from native forests compared to the grassland and timber plantation samples (Fig. 4l-o and 5l-o; Table A1-A4). At the same time, samples from *Eucalyptus* plantations had higher levels of both Cr ($p<0.005$) and Pb ($p<0.005$) than *Pinus* topsoils, while the same was observed for Cr ($p<0.001$) and Pb ($p<0.05$) in riverine forests compared to park forests (Fig. S5).

# 4. Discussion

The vicious circle between the wish to stop soil degradation and concurrent increases in land productivity to satisfy the increasing demand for food, fibres and energy has not been broken since green revolution. Socio-economic and conventional management practices that drive soil degradation have generated several traps, such as the 'inputs trap' where a reduced yield per area is followed by higher fertilizer application or the 'credit or poverty trap' where economic pressure forces the famer to intensification (Gomiero, 2016). Caught in this loop, the soils of the temperate grasslands of Uruguay have suffered strong degradation from erosion, acidification, contamination, salinification and compaction (Zurbriggen et al., 2020). This is clearly reflected in the results of our topsoil survey, which also adds interesting insights from timber plantations, grasslands and native forests to an existing database consisting mainly of crops and pastures samples from 2002-2014, which demonstrated the loss of organic matter by 25% and an increasing loss of nutrients (Beretta-Blanco et al. 2019). We contribute deeper insights on fertility, acidification and trace metals accumulation in topsoils from a wide range of different land uses, which is, to our knowledge, unique for the region since the CONEAT classification (CONEAT Index, 1976).



## 4.1 Translocation of elements in topsoils within landscape

Our data demonstrate the accumulation of SOM, nutrients and trace metals in topsoil samples from riverine forests, suggesting transport of soil particles from the surrounding land uses (e.g., grasslands, crop or timber plantations) to the borders of rivers, streams and creeks. Regional soil erosion models estimate the loss of 2-5 tons ha$^{-1}$ year$^{-1}$ for a third of the country depending on precipitation, topography, soil erodibility and land management (Carrasco-Letelier and Beretta-Blanco 2017). One possible direct impact is the increasing eutrophication reported for larger local rivers, although the models used by these authors did not link Chlorophyll-a concentrations with agricultural land use (Beretta-Blanco and Carrasco-Letelier 2021 and replies).

Organic matter content and the exchangeable cations are strongly reduced in topsoils of grasslands, timber plantations and crops compared to native forests (Fig. 4b, d-h and 5b, d-h). Of all the fertility proxies assessed, phosphorus in topsoil was most significantly affected by different land uses, being highest in native forests (Fig. 4c and 5c). The cation exchange capacity (CEC) was highest in native forests (more than 160 percent of the CEC in grasslands) and lowest in timber plantations, reaching only the half of the CEC in grasslands (Table A1-A2). Lower average nutrient contents and corresponding CEC reported for two timber plantations in the East and one in the Norwest of Uruguay (Hernández et al., 2009 and 2016; Céspedes-Payret et al., 2012) may result from a combined effect of topsoil degradation in timber plantations due to management practices and to soil texture (Sandoval-Lopez et al., 2020). The trees' uptake and the general export of nutrients from fast growing timber plantations through harvesting is higher than the natural input into those systems (Merino et al., 2005). This effect is particularly relevant because timber plantations in Uruguay are on 'forestry priority soils', which are generally soils with low fertility, superficial to moderate depth and good drainage (OAS, 1994), so afforestation might reduce even more their soil fertility.




Comparing neighbouring sites of grassland and afforested grasslands, our data set shows no significant depletion of nutrients by timber plantations (p=0.208) but a slightly higher average of CEC at grasslands (Table A1). In general, plants are expected to compensate the extraction of cations from upper soil by the 'uplift of nutrient' from deeper horizons (Jobbagy and Jackson, 2004). This has been questioned for *Eucalyptus* plantations on sandy soils, where, for example, an increase of potassium in the topsoil was not observed compared to the neighbouring grassland (Céspedes-Payret et al., 2012). Phosphorous and cations decline with sand content, so physicochemical factors such as the percentage of sand and organic matter, influence soil fertility (Sandoval-Lopez et al., 2020). A study of *Eucalyptus* plantations in South-East Brazil did not show a significant depletion of nutrients and carbon; in contrast, carbon, potassium and calcium increased in the topsoil after twelve years over one or two harvest and fertilization cycles (McMahon et al., 2019). To sum up, the patterns observed in the various sites in our survey and the other studies indicate very complex interactions of numerous factors. Removal of nutrients by high-yield timber plantations may exceed the capacity of nutrient exchange or turnover of forest litter and wastes. Both stand management and environmental conditions (e.g., precipitation) influence nutrient and carbon stocks (McMahon et al., 2019; Sandoval-Lopez et al., 2020).

The C/N ratio in topsoils ranged within values reported for grassland and timber plantations of the region (see Berthrong et al., 2012). In contrast to that study, however, we observed no differences between grassland and timber plantations (Fig. 4a), but between topsoils of timber plantations, native forests and crops (Fig. 5a). Organic matter and nitrogen decrease in topsoils after grassland afforestation (Berthrong et al., 2009) and C/N ratio increases with plantation age and decreases with precipitation (Berthrong et al., 2012). The topsoils of timber plantations have, on average, lower N contents compared to other land uses (Table A2) and data in literature (Jobbagy and Jackson 2003).



Although in cropland, nutrients are regularly compensated for by increased application of
fertilizers (Beretta-Blanco et al., 2019), timber plantations are usually fertilized only in the first
year of planting (e.g., Binkley et al., 2017; Sandoval-Lopez et al., 2020). The CEC and the
average ratio between $K^+(Ca^{2+} +Mg^{2+})^{-1/2}$ in crops and grasslands are in the ranges reported by
Beretta-Blanco et al. (2019) with lower availability of potassium for crops. A shortage of
potassium and calcium is also most likely for future timber plantations after harvest, especially
if the logging process do not include bark stripping on site (Hernández et al., 2009) and due to
short time spans between harvest and new planting in the same area (Sandoval-Lopez et al.,
2020), but see contrasting results of McMahon et al. (2019). The extraction of nutrients and
biomass due to grazing (Fernandez et al., 2017), timber (Hernández et al., 2009 and 2016) and
crop production (Beretta-Blanco et al., 2019) lowers organic matter content and exchangeable
cations, affects the physical, chemical and biological soil properties, and drives degradation.

## 4.2 Acidification in Uruguayan topsoils across land uses

A further dimension of the soil degradation directly linked to the cation extraction is the
acidification of soils. This has been demonstrated for crops and pastures (Beretta-Blanco et al.,
2019), and a study site with *Eucalyptus* (Céspedes-Payret et al., 2012) and is now broadly
supported by our topsoil samples originating from a wide range of different land uses across
Uruguay. The pH values of our topsoil samples are mainly in the category of very strongly to
extremely acidic with lowest values for timber plantations (Fig. 6), which are below the means
reported so far (Jobbagy and Jackson, 2003; Céspedes-Payret et al., 2012). Moreover, as our
data on topsoil pH fall short compared to the values estimated by the Food and Agriculture
Organization of the United Nations and the Intergovernmental Technical Panel on Soils (Caon
and Vargas, 2017), acidification and the deterioration of topsoil quality continues.
Acidification results from intensified land uses with nitrogen fertilization, with biological N
fixation by the legumes both used in the so-called "improved pastures" (Modernel et al., 2016),
or with cation extraction by crop or timber harvest (Jobbagy and Jackson, 2003). So far,

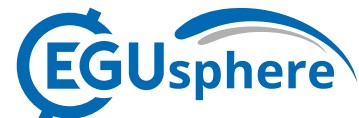

although forest soils tend to be more acidic than agricultural soils due to acid-neutralizing
treatments in the latter (Baize and van Oort, 2014), we found no differences between topsoils
of crops and native forests (Fig. 6d) as the high organic matter content in native forests buffers
the process to a certain extent.

## 4.3 Riverine Forest soils as sink for trace metals

In general, the average concentration of Zn, Cu, Cd, Cr, As and Pb in our topsoils were within
the expected ranges of samples from crops, pastures and grassland in the region (Lavado et al.,
1998, 2004; Roca, 2015). Some samples, especially from orchards and crops, exceeded the
background values of copper, cadmium and arsenic (Table A2; Roca, 2015). Little is known
about the pedo-geochemical background of the Pampean soils (Roca, 2015): for example, high
arsenic content in Uruguayan ground waters has been hypothesized to be due to quaternary ash
deposits (Wu et al., 2021). In addition, the lateral heterogeneity of Pampean soils over short
distances makes separating geochemical and anthropic signatures difficult (Roca, 2015).
However, that the main risk of soil contamination in the region is from the application of
fertilizers and agrochemicals is uncontested (Kaushal et al., 2018).
To our knowledge, there has been no regional study of trace metals in the native riverine forests
or timber plantations. Our work thus expands the evidence base for these land uses. The topsoils
of riverine forests accumulate trace metals compared to those of timber plantations and crops
(Fig. 4l-o). The higher amount of soil organic matter in riverine forests favours the retention of
cations, including trace metals. Although the origin of potentially harmful elements in forest
soils have been primarily attributed to atmospheric deposition (Baize and van Oort, 2014),
atmospheric deposition only plays a major role in vicinity of urban or industrial development,
and our data from rural sites suggests a different entry path from the surrounding land uses to
the riverine forests. High levels of acidification and low amounts of organic matter reduce the
retention of trace metals in the soil of timber plantations, and elements leach out of the soil
towards the water table (Baize and van Oort, 2014). The acidification strongly contributes to



the overall mobility of base cations into the 'chemical cocktail of the Anthropocene' (Kaushal
et al., 2018), including trace metals. We thus observe positive feedback in already
impoverished soils with high acidity favouring cations solubility, in addition to the uptake by
trees intensifies this effect. Timber plantations extract trace metals from soil and also
accumulate it in the bark or leaves, so they have been used for phytoremediation (Li et al.,
2020). This may explain the higher values of cadmium in grassland compared to timber
plantations (Fig. 4m). Differences between *Eucalyptus* and *Pinus* stands may be related to
different age classes, as the later may have extracted more lead and chromium from the soil
due to their older stand age with rotation periods of about 20 years (Li et al., 2020).

### 4.4 Carbon storage in topsoils of *Eucalyptus* plantations?

Our study provides evidence that the loss of soil organic matter limits not only the productivity
of the crops, but also potential carbon sequestration in the region (Beretta-Blanco et al., 2019).
Grassland conversion to cropland decreases soil organic carbon storage compared to adjacent
native grasslands (Hernandez-Ramirez et al., 2021).
Afforestation of croplands has been also discussed as a carbon sequestration measure to
proactively address and effectively mitigate ongoing climate change within a person's lifetime
(Hernandez-Ramirez et al., 2021; Mayer et al. 2020). Although carbon stocks of four
*Eucalyptus* stands in the Brazilian Cerrados increased but did not change in four *Eucalyptus*
stands within the Atlantic Forest (McMahon et al., 2019), this may not be the case for short
rotation of *Eucalyptus* plantation on Campos grasslands. In our topsoil samples originating
from 28 different stands across Uruguay, organic matter is lowest in topsoils of timber
plantations (Fig. 4b). Similar amounts have been reported for a timber plantation in the East of
Uruguay (Céspedes-Payret et al., 2012), and in North-Eastern Argentine (Sandoval-Lopez et
al., 2020). Our data therefore provide clear evidence that rather than contributing to carbon
sequestration in the topsoil, the carbon release from the transformation of native grasslands to



plantation with these fast-growing species has several adverse effects depending on
precipitation and soil type (reviewed by Mayer et al., 2020).
Several trade-offs between carbon sequestration through afforestation and local water yield and
soil fertility have been demonstrated, including nutrient and soil organic matter depletion,
acidification, and biodiversity loss and corresponding challenges for landscape conservation
(e.g., Jackson et al., 2005; Veldman et al., 2015; Friggens et al., 2020). Soil carbon changes
dynamically during the first decade of afforestation. Remaining grassland carbon declines,
while tree carbon gain starts (e.g. Paul et al., 2002). Although a net gain of carbon is expected
when the new forests approached equilibrium after decades (Hernández-Ramirez et al., 2021),
in contrast to long-lasting forests, *Eucalyptus* harvest in Uruguay takes place after less than a
decade (appr. 7-10 years). Soil organic matter does not differ between *Eucalyptus* plantations
and neighbouring grasslands (Appendix A; Table A1 and A2). Another study near our sites 12
and 13 also did not find a significant difference between the soil organic carbon of the upper
soil (0-30cm) compared to afforested versus native grasslands (Hernández et al., 2016). A study
near our sites 7, 8, 10 reported higher top soil carbon of grassland compared with afforested
sites in their vicinity only if sand content is lower than sixty percent (Sandoval-Lopez et al.,
2020). McMahon et al. (2019) identified a greater carbon gain under *Eucalyptus* stands
compared to (potential) carbon losses in neighbouring degraded Cerrado grasslands. There is
evidence that the retention of residues after harvest increases the carbon stock in soil (Mayer
et al., 2020). Long term monitoring of carbon stocks is needed to verify the influences of
management and environmental changes. A regional study on *Eucalyptus* plantations across
different biomes in Brazil shows both decreases and no changes depending on precipitation in
the dry season, clay content and on the initial stock of carbon in the soil (Cook et al., 2016)
The simplistic solution of huge tree plantations to compensate anthropogenic $CO_2$ emissions
has been challenged in the last decade, and some crucial lessons learnt have been summarized
(Di Sacco et al., 2020). Since grasslands are more sustainable carbon sinks compared to





climate-vulnerable monocultures such as timber plantations (Dass et al. 2018), avoiding
afforestation of previously non-forested lands is important. This is the case for *Eucalyptus*
afforestation in the originally forestless grasslands of Uruguay: models on carbon sequestration
and dynamics in Mollisols and Oxisols under South American grasslands estimated a higher
carbon retention efficiency under grasslands compared to afforested sites, suggesting that
silvopastoral systems are a potential solution for soil carbon sequestration in tropical soils
(Berhongaray and Alvarez, 2019).
Connecting or expanding existing forest and using native species for plantings (Di Sacco et al.,
2020) is also recommended (see also Alvaro et al. 2020, Ortiz et al. 2020). Our topsoil data
indicates that carbon sequestration occurs mainly in the topsoils of native riverine forests that
cover less than five percent of the Uruguayan territory. Consequently, the expansion of native
forests and the use of native species in forestry project for long term establishment can reduce
adverse effects of timber plantations. While there is preliminary evidence that N-fixing tree
species can help increase local C stocks of afforestation, because of their potential for invasion,
exotic N-fixers should be avoided (Mayer et al. 2020).
**4.5. Back to more conservative grassland management?**
Our soil survey data shows strong soil degradation of Uruguayan black soils from erosion,
acidification and contamination, and indicates a translocation of nutrients and organic matter
across the landscape from grassland, timber and crop plantations to the riverine forests. The
potential of grasslands as cropland reserve have been largely overestimated (Lambin et al.,
2013), and they have already degraded during the last decades by inappropriate land
management techniques (Jaurena et al., 2021, De Faccio et al., 2021), lack of mainstreaming
soil conservative techniques (García-Préchac, 2004; Fernandez et al., 2017), decoupling of crop
and lifestock (De Faccio et al, 2021) plus climate change impacts with storm water events and
drought, all of which trigger soil erosion (Wingeyer et al. 2015). From the although very limited
point of view on topsoils, the concept of conserving 'old growth grasslands' with extensive use

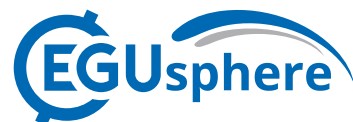

(Veldman et al. 2015) appears a more promising strategy to put the 'grasslands at the core' in
the Campos region than the use intensification strategies envisioned by Jaurena et al. (2021).

## Data availability

All data generated or analysed in this study are included in this article (and its supplementary
materials). Further data are available from the corresponding authors upon reasonable request.

## Author contributions

IS, LRR, ST and MB conceptualized the study, performed the soil sampling and data analysis.
EZ advised the laboratory analyses. IS wrote the initial draft. LRR, ST, MB and EZ contributed
to generating and reviewing the subsequent versions of the manuscript. IS received the funding
for the study.

## Competing interests

The authors declare that they have no conflict of interest

## Acknowledgement

In alphabetical order, we thank Juan Barreneche, Lucia Gaucher, Sören Miehe, Nicolas Silvera
and Matias Zarucki for their assistance in field work. We thank Manuel Garcia and Meica
Valdivia for the pre-processing of samples. We thank the staff of the soil lab from the
Department of Soil and Natural Resources, University of Concepción, Chile. We thank Diego



de Panis for help with statistical analysis and visualization and valuable comments on a
previous version of this manuscript. Thanks to Vera Krause, Serafina Bischoff, Sophia Reitzug,
Rhea Rennert and Diego Nicolas Rojas for support with data analysis and plots and maps
visualization. We also thank all landowners for access permission to establish our monitoring
sites on their land, their hospitality and willingness to discuss land use goals concerning all
dimension of sustainability. The study was funded by the German Federal Ministry of
Education and Research (BMBF; 01LN1305A). Special thanks go to Amal Chatterjee for
improving our English.

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





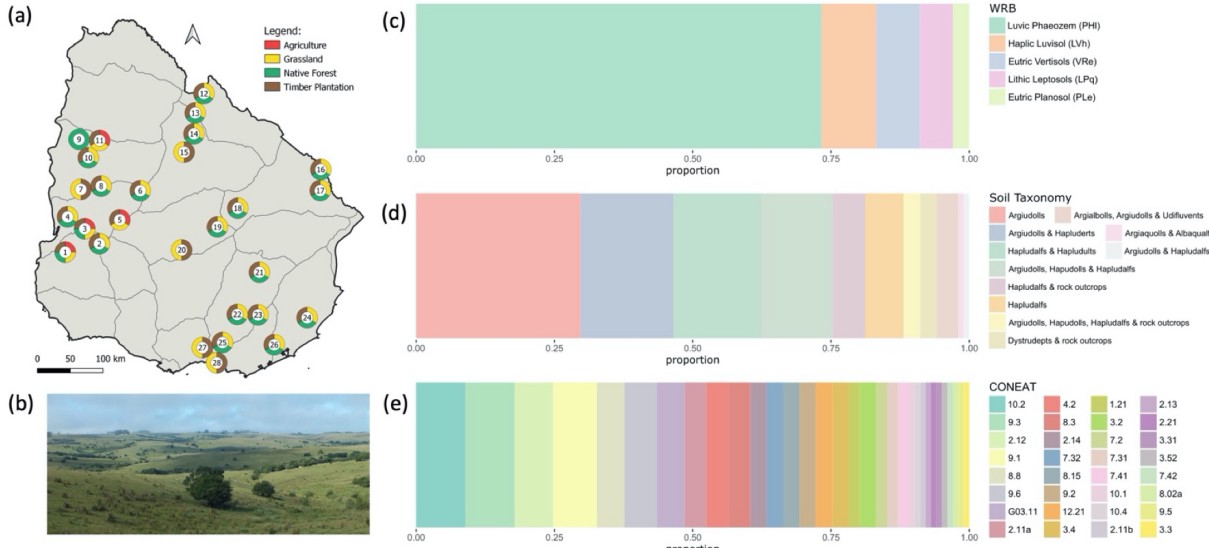

**Figure 1:** Study design and sampling. (a) Location of the 28 monitoring sites across Uruguay including land use types sampled (grassland (b); timber plantations, native forest and agricultural land. Proportion of plots with particular category of soil classification according to the World Reference Base (WRB, c), Soil Taxonomy (d) and CONEAT (Comisión Nacional de Estudio Agronómico de la Tierra, e).




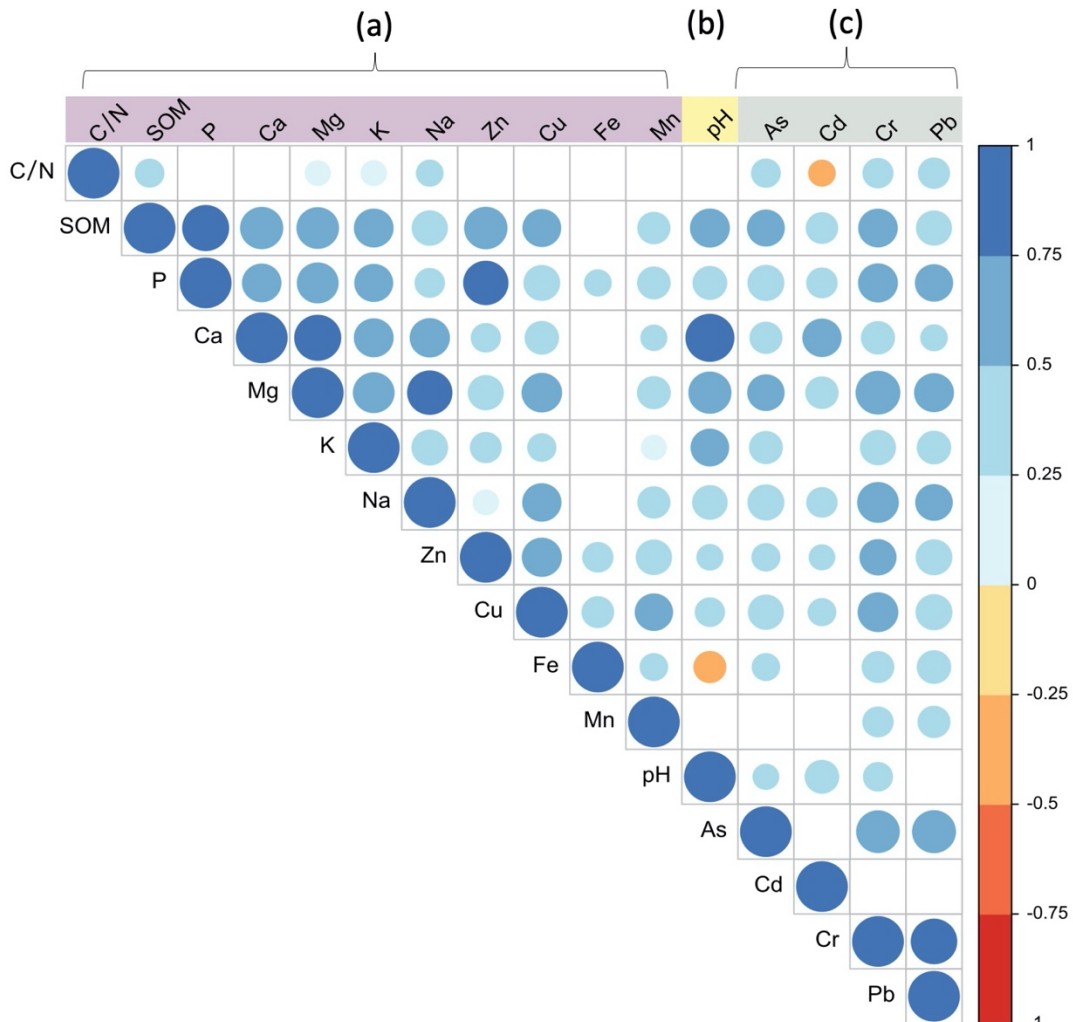

**Figure 2.** Spearman's rank correlations for parameters of topsoils (n=80) regarding (a) fertility proxies, (b) acidity and (c) trace metals. Colour intensity and the size of the circle are proportional to the correlation coefficients ($\rho$). Empty slots show correlations with p>0.05.









**Figure 3.** Non-metric multidimensional scaling showing significant clustering differences
among samples from (a) grassland (GL), timber plantations (TP) and native forests (NF); (b)
among samples from *Pinus* and *Eucalyptus* plantations; (c) among land uses (including
agriculture AC) in Argiudolls; (d) among land uses in Argiudolls & Hapluderts; (e) among land
uses in Argiudolls, Hapudolls & Hapludalf and (f) among land uses in Hapludalfs &
Hapludults.









**Figure 4.** Violin box plots for significant Kruskal-Wallis Tests across evaluated land uses (i.e.
grassland (GL), timber plantation (TP) and native forest (NF)) for each variable. Significance
in posterior Wilcoxon pairwise comparisons is depicted as $p < 0.05$ (*), $p < 0.01$ (**), $p < 0.001$
(***), $p < 0.0001$ (****).












**Figure 5**. Violin box plots for significant Kruskal-Wallis Tests in 'Argiudolls' Soil Taxonomy category for fertility variables across available land uses (GL: Grassland, TP: Timber plantation, NF: Native forest, AC: Agriculture). Significance in posterior Wilcoxon pairwise comparisons is depicted as p<0.05 (*), p<0.01 (**), p<0.001 (***), p<0.0001 (****).

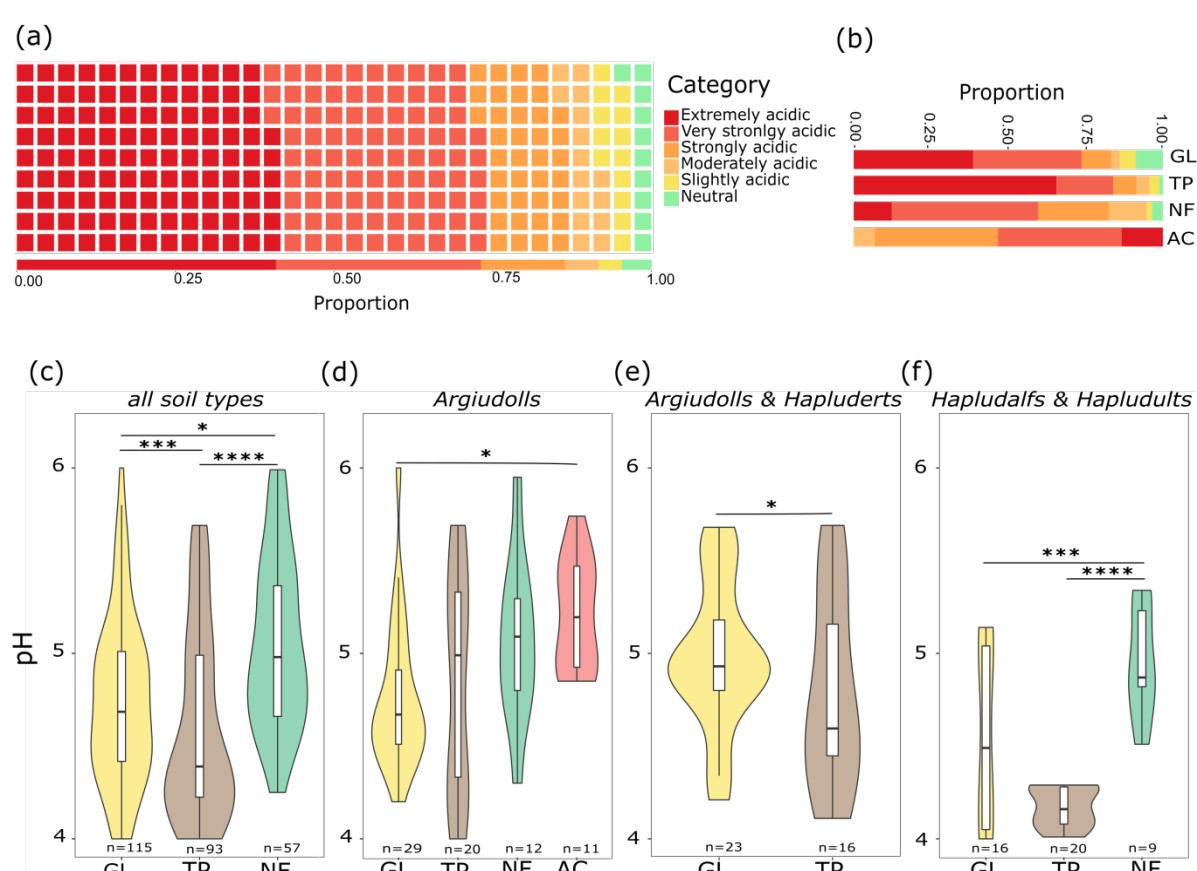

**Figure 6.** Acidity of all topsoils samples (a) and (b) according to different land use (GL: Grassland, TP: Timber plantation, NF: Native forest, AC: Agriculture). Proportion of samples with a given acidic category (Kellogg, 1993). pH of topsoil given as violin box plots for



significant Kruskal-Wallis Tests across different land uses independent from soil Order (c) and
separated per soil Orders (d-f). Sample number (n) is given. Significance in posterior Wilcoxon
pairwise comparisons is depicted as $p<0.05$ (*), $p<0.01$ (**), $p<0.001$ (***), $p<0.0001$ (****).





**Table 1.** General characteristics of Uruguayan topsoils: descriptive statistics for the parameters of all single samples (n) across different land uses and classification to different soil types (for details on different land uses i.e. grassland, timber plantations, native forests and crops see Appendix A1-A2 and on different soil types see Appendix A3-4).

| Variable | n | Mean | S.D. | Minimum | Maximum |
|---|---|---|---|---|---|
| Humidity (%) | 280 | 18.9 | 10.8 | 1.34 | 51.5 |
| N_total (%) | 280 | 0.2 | 0.14 | 0.04 | 1.2 |
| C_total (%) | 280 | 3.0 | 2.3 | 0.4 | 25.7 |
| C/N ratio | 279 | 13.3 | 1.8 | 9.1 | 20.6 |
| SOM (%) | 267 | 5.6 | 3.0 | 0.8 | 16.4 |
| P (mg kg$^{-1}$) | 278 | 295 | 166 | 43 | 1008 |
| Ca (cmol kg$^{-1}$) | 82 | 9.4 | 8.9 | 0.3 | 42.7 |
| Mg (cmol kg$^{-1}$) | 82 | 2.5 | 2.2 | 0.1 | 9.7 |
| K (cmol kg$^{-1}$) | 82 | 0.4 | 0.25 | 0.03 | 1.44 |
| Na (cmol kg$^{-1}$) | 82 | 0.1 | 0.14 | 0 | 0.69 |
| Zn (mg kg$^{-1}$) | 82 | 1.9 | 2.1 | 0.1 | 9.7 |
| Cu (mg kg$^{-1}$) | 82 | 5.4 | 22.1 | 0.3 | 161.2 |
| Fe (mg kg$^{-1}$) | 82 | 134 | 75 | 5 | 309 |
| Mn (mg kg$^{-1}$) | 81 | 25 | 17 | 0.7 | 93 |
| pH | 279 | 4.8 | 0.8 | 3.6 | 7.34 |
| As (mg kg$^{-1}$) | 279 | 3.9 | 3.6 | 0.6 | 30.7 |
| Cd (mg kg$^{-1}$) | 274 | 0.4 | 0.2 | 0.2 | 1 |
| Cr (mg kg$^{-1}$) | 280 | 13.6 | 12.6 | 2 | 78.9 |
| Pb (mg kg$^{-1}$) | 280 | 7 | 3.4 | 2 | 27.6 |
| CEC (cmol kg$^{-1}$) | 82 | 12.4 | 10.7 | 0.5 | 50.1 |
| $K^+/(Ca^{2+}+Mg^{2+})$ | 82 | 0.07 | 0.08 | 0.01 | 0.43 |
| $K^+/(Ca^{2+}+Mg^{2+}+Na^+)$ | 82 | 0.07 | 0.07 | 0.01 | 0.42 |





# Appendices

## Appendix A

**Table A1.** Descriptive statistics of topsoil variables for all single soil types at grassland and native forests. Number of samples (n), mean, standard deviation (SD), minimum (Min) and maximum (Max) of each variable are given.

| | Grassland | | | | | Native forests | | | | |
|---|---|---|---|---|---|---|---|---|---|---|
| Variable (unit) | n | Mean | SD | Min | Max | n | Mean | SD | Min | Max |
| Humidity (%) | 115 | 20.4 | 11.5 | 1.5 | 51.5 | 57 | 25.8 | 9.3 | 6.3 | 50.3 |
| N_total (%) | 115 | 0.22 | 0.13 | 0.06 | 0.80 | 57 | 0.30 | 0.19 | 0.04 | 1.20 |
| C_total (%) | 115 | 2.9 | 1.9 | 0.7 | 10.8 | 57 | 4.2 | 3.6 | 0.5 | 25.7 |
| C/N ratio | 115 | 13.2 | 1.7 | 9.1 | 19.4 | 56 | 13.5 | 1.4 | 9.8 | 18.5 |
| SOM (%) | 109 | 5.3 | 2.6 | 1.4 | 15.2 | 53 | 7.5 | 3.8 | 1.1 | 16.4 |
| P (mg/kg) | 114 | 272 | 124 | 43 | 749 | 57 | 484 | 194 | 56 | 1009 |
| Ca (cmol kg$^{-1}$) | 32 | 9.5 | 10.0 | 0.3 | 42.7 | 18 | 15.1 | 10.0 | 0.3 | 37.3 |
| Mg (cmol kg$^{-1}$) | 32 | 2.2 | 2.1 | 0.18 | 9.7 | 18 | 4.8 | 2.6 | 0.12 | 8.9 |
| K (cmol kg$^{-1}$) | 32 | 0.40 | 0.26 | 0.08 | 1.44 | 18 | 0.50 | 0.28 | 0.03 | 1.29 |
| Na (cmol kg$^{-1}$) | 32 | 0.12 | 0.15 | 0.00 | 0.65 | 18 | 0.14 | 0.12 | 0.00 | 0.50 |
| Zn (mg kg$^{-1}$) | 32 | 1.4 | 1.2 | 0.10 | 4.7 | 18 | 4.0 | 2.7 | 0.80 | 9.60 |
| Cu (mg kg$^{-1}$) | 32 | 2.2 | 1.8 | 0.30 | 7.7 | 18 | 2.4 | 1.5 | 0.30 | 5.50 |
| Fe (mg kg$^{-1}$) | 32 | 121 | 81 | 5 | 279 | 18 | 149 | 76 | 10 | 309 |
| Mn (mg kg$^{-1}$) | 32 | 22 | 18 | 2 | 93 | 17 | 33 | 18 | 1 | 72 |
| pH | 114 | 4.9 | 0.9 | 3.6 | 7.3 | 57 | 5.1 | 0.7 | 3.6 | 7.2 |
| As (mg kg$^{-1}$) | 115 | 4.3 | 4.5 | 0.6 | 30.7 | 56 | 4.1 | 2.5 | 1.2 | 13.6 |
| Cd (mg kg$^{-1}$) | 111 | 0.41 | 0.17 | 0.17 | 1.00 | 57 | 0.45 | 0.15 | 0.17 | 0.81 |
| Cr (mg kg$^{-1}$) | 115 | 12.3 | 9.2 | 2.6 | 49.5 | 57 | 22.8 | 17.3 | 2.0 | 70.7 |
| Pb (mg kg$^{-1}$) | 115 | 6.8 | 3.0 | 2.3 | 15.9 | 57 | 9.5 | 4.8 | 2.9 | 27.7 |
| CEC (cmol/ kg$^{-1}$) | 32 | 12.2 | 11.5 | 0.8 | 50.1 | 18 | 20.4 | 11.4 | 0.5 | 44.6 |
| $K^+/(Ca^{2+}+Mg^{2+})$ | 32 | 0.08 | 0.10 | 0.01 | 0.43 | 18 | 0.03 | 0.02 | 0.01 | 0.07 |
| $K^+/(Ca^{2+}+Mg^{2+}+Na^+)$ | 32 | 0.08 | 0.10 | 0.01 | 0.42 | 18 | 0.03 | 0.02 | 0.01 | 0.07 |



**Table A2.** Descriptive statistics for of topsoil variables for all single soil types at timber
plantations and crops. Number of samples (n), mean, standard deviation (SD), minimum (Min)
and maximum (Max) of each variable are given.

| Variable (unit) | Timber plantations | | | | | Crops | | | | |
|---|---|---|---|---|---|---|---|---|---|---|
| | n | Mean | SD | Min | Max | n | Mean | SD | Min | Max |
| Humidity (%) | 93 | 13.2 | 7.0 | 1.7 | 38.0 | 15 | 18.1 | 12.0 | 1.3 | 50.5 |
| N_total (%) | 93 | 0.18 | 0.10 | 0.04 | 0.61 | 15 | 0.22 | 0.06 | 0.13 | 0.29 |
| C_total (%) | 93 | 2.5 | 1.5 | 0.4 | 8.4 | 15 | 2.6 | 0.7 | 1.4 | 3.3 |
| C/N ratio | 93 | 13.6 | 2.1 | 10.0 | 20.6 | 15 | 11.7 | 0.7 | 10.5 | 12.8 |
| SOM (%) | 90 | 4.9 | 2.8 | 0.8 | 16.1 | 15 | 5.2 | 1.3 | 2.8 | 6.7 |
| P (mg kg$^{-1}$) | 93 | 206 | 91 | 56 | 551 | 14 | 310 | 120 | 142 | 613 |
| Ca (cmol kg$^{-1}$) | 27 | 5.2 | 4.8 | 0.6 | 18 | 5 | 10.8 | 5.5 | 7.2 | 20.4 |
| Mg (cmol kg$^{-1}$) | 27 | 1.4 | 1.0 | 0.16 | 4.1 | 5 | 2.5 | 1.0 | 1.76 | 4.1 |
| K (cmol kg$^{-1}$) | 27 | 0.32 | 0.21 | 0.09 | 0.93 | 5 | 0.46 | 0.23 | 0.29 | 0.85 |
| Na (cmol kg$^{-1}$) | 27 | 0.11 | 0.16 | 0.00 | 0.69 | 5 | 0.06 | 0.02 | 0.03 | 0.08 |
| Zn (mg kg$^{-1}$) | 27 | 0.94 | 0.74 | 0.10 | 2.8 | 5 | 3.7 | 3.7 | 0.60 | 9.7 |
| Cu (mg kg$^{-1}$) | 27 | 1.5 | 0.9 | 0.5 | 4.5 | 5 | 57.7 | 78.8 | 0.8 | 161.0 |
| Fe (mg kg$^{-1}$) | 27 | 143 | 69 | 16 | 290 | 5 | 115 | 62 | 63 | 210 |
| Mn (mg kg$^{-1}$) | 27 | 25 | 16 | 6 | 72 | 5 | 17 | 5 | 10 | 23 |
| pH | 93 | 4.5 | 0.7 | 3.6 | 6.8 | 15 | 5.1 | 0.4 | 4.2 | 5.7 |
| As (mg kg$^{-1}$) | 93 | 3.4 | 3.0 | 0.7 | 25.5 | 15 | 3.1 | 1.2 | 1.1 | 5.5 |
| Cd (mg kg$^{-1}$) | 91 | 0.33 | 0.11 | 0.16 | 0.65 | 15 | 0.41 | 0.14 | 0.21 | 0.66 |
| Cr (mg kg$^{-1}$) | 93 | 10.6 | 11.3 | 2.0 | 78.9 | 15 | 7.8 | 1.8 | 5.0 | 11.3 |
| Pb (mg kg$^{-1}$) | 93 | 6.1 | 2.3 | 2.0 | 11.7 | 15 | 5.1 | 1.2 | 3.4 | 7.3 |
| CEC (cmol kg$^{-1}$) | 27 | 7.0 | 5.6 | 1.0 | 20.6 | 5 | 13.8 | 6.5 | 9.5 | 25.0 |
| K$^{+}$/(Ca$^{2+}$+Mg$^{2+}$) | 27 | 0.08 | 0.07 | 0.02 | 0.25 | 5 | 0.04 | 0.03 | 0.02 | 0.09 |
| K$^{+}$/(Ca$^{2+}$+Mg$^{2+}$+Na$^{+}$) | 27 | 0.08 | 0.07 | 0.02 | 0.25 | 5 | 0.04 | 0.03 | 0.02 | 0.08 |






**Table A3.** Descriptive statistics of topsoil variables for Argiudolls and for grassland and native
forests within of Argiudolls. Number of single samples (n), mean, standard deviation (SD),
minimum (Min) and maximum (Max) of each variable are given.

| Variable (Unit) | Total Argiudolls | | | | | Grassland | | | | | Native forests | | | | |
|---|---|---|---|---|---|---|---|---|---|---|---|---|---|---|---|
| | n | Mean | SD | Min | Max | n | Mean | SD | Min | Max | n | Mean | SD | Min | Max |
| Humidity (%) | 77 | 18.3 | 10.9 | 1.3 | 50.5 | 31 | 19.0 | 9.2 | 5.0 | 49.0 | 12 | 26.4 | 13.4 | 9.0 | 48.2 |
| N_total (%) | 77 | 0.24 | 0.15 | 0.07 | 1.20 | 31 | 0.21 | 0.11 | 0.07 | 0.66 | 12 | 0.34 | 0.28 | 0.13 | 1.20 |
| C_total (%) | 77 | 3.3 | 3.0 | 1.1 | 25.7 | 31 | 2.9 | 1.8 | 1.1 | 10.8 | 12 | 5.3 | 6.5 | 1.7 | 25.7 |
| C/N ratio | 76 | 13.3 | 2.2 | 9.8 | 19.9 | 31 | 13.3 | 2.1 | 10.5 | 19.4 | 11 | 13.6 | 2.1 | 9.8 | 18.5 |
| SOM (%) | 73 | 5.8 | 2.5 | 2.1 | 16.1 | 30 | 5.2 | 1.9 | 2.1 | 10.0 | 9 | 7.4 | 2.1 | 4.8 | 11.0 |
| P (mg kg$^{-1}$) | 76 | 280 | 122 | 93 | 693 | 30 | 234 | 78 | 93 | 353 | 12 | 438 | 155 | 155 | 693 |
| Ca (cmol kg$^{-1}$) | 21 | 11.4 | 8.0 | 2.6 | 37.1 | 8 | 11.5 | 6.7 | 3.2 | 20.5 | 4 | 16.2 | 14.3 | 5.1 | 37.1 |
| Mg (cmol kg$^{-1}$) | 21 | 2.3 | 1.4 | 0.5 | 6.1 | 8 | 2.0 | 0.8 | 0.9 | 3.2 | 4 | 3.5 | 2.5 | 1.0 | 6.1 |
| K (cmol kg$^{-1}$) | 21 | 0.45 | 0.26 | 0.1 | 1.3 | 8 | 0.4 | 0.14 | 0.16 | 0.61 | 4 | 0.69 | 0.41 | 0.34 | 1.29 |
| Na (cmol kg$^{-1}$) | 21 | 0.08 | 0.05 | 0.03 | 0.26 | 8 | 0.07 | 0.02 | 0.04 | 0.10 | 4 | 0.10 | 0.11 | 0.03 | 0.26 |
| Zn (mg kg$^{-1}$) | 21 | 2.4 | 3.0 | 0.1 | 10 | 8 | 1.3 | 0.8 | 0.1 | 2.2 | 4 | 5.1 | 4.3 | 1.3 | 10 |
| Cu (mg kg$^{-1}$) | 21 | 15.5 | 42.7 | 0.8 | 161 | 8 | 2.6 | 1.6 | 1.0 | 6.0 | 4 | 1.9 | 1.0 | 0.8 | 3.0 |
| Fe (mg kg$^{-1}$) | 21 | 117 | 62 | 10 | 232 | 8 | 104 | 50 | 28 | 171 | 4 | 117 | 84 | 10 | 216 |
| Mn (mg kg$^{-1}$) | 21 | 23.3 | 10.4 | 7.1 | 43.6 | 8 | 20.5 | 10.4 | 7.1 | 33.8 | 4 | 28.8 | 10.4 | 15.4 | 39.4 |
| pH | 77 | 5.0 | 0.7 | 3.8 | 6.6 | 31 | 4.9 | 0.7 | 3.9 | 6.6 | 12 | 5.2 | 0.5 | 4.3 | 6.1 |
| As (mg kg$^{-1}$) | 76 | 3.2 | 1.4 | 1.0 | 12.2 | 31 | 2.9 | 0.9 | 1.5 | 4.9 | 11 | 4.0 | 2.8 | 1.5 | 12.2 |
| Cd (mg kg$^{-1}$) | 76 | 0.4 | 0.1 | 0.2 | 0.7 | 30 | 0.4 | 0.1 | 0.2 | 0.7 | 12 | 0.4 | 0.1 | 0.2 | 0.7 |
| Cr (mg kg$^{-1}$) | 77 | 9.3 | 6.2 | 2.2 | 28.3 | 31 | 8.1 | 4.1 | 3.6 | 18.5 | 12 | 13.2 | 9.5 | 2.2 | 28.3 |
| Pb (mg kg$^{-1}$) | 77 | 5.6 | 2.1 | 2.8 | 10.7 | 31 | 5.2 | 1.8 | 2.8 | 9.6 | 12 | 6.8 | 2.9 | 2.9 | 10.7 |
| CEC (cmol kg$^{-1}$) | 21 | 14.2 | 9.3 | 3.4 | 44.6 | 8 | 14.0 | 7.3 | 4.5 | 23.1 | 4 | 20.4 | 16.7 | 6.4 | 44.6 |
| K$^{+}$/(Ca$^{2+}$+Mg$^{2+}$) | 21 | 0.04 | 0.02 | 0.02 | 0.09 | 8 | 0.03 | 0.02 | 0.02 | 0.07 | 4 | 0.04 | 0.01 | 0.03 | 0.06 |
| K$^{+}$/(Ca$^{2+}$+Mg$^{2+}$+Na$^{+}$) | 21 | 0.04 | 0.02 | 0.02 | 0.08 | 8 | 0.03 | 0.02 | 0.02 | 0.07 | 4 | 0.04 | 0.01 | 0.03 | 0.06 |





**Table A4.** Descriptive statistics of topsoil variables for Argiudolls and timber plantations and
crops within of Argiudolls. Number of single samples (n), mean, standard deviation (SD),
minimum (Min) and maximum (Max) of each variable are given.

| Variable (unit) | | Total Argiudolls | | | | | Timber plantations | | | | | Crops | | | |
|---|---|---|---|---|---|---|---|---|---|---|---|---|---|---|---|
| | n | Mean | SD | Min | Max | n | Mean | SD | Min | Max | n | Mean | SD | Min | Max |
| Humidity (%) | 77 | 18.3 | 10.9 | 1.3 | 50.5 | 22 | 14.0 | 7.8 | 5.5 | 38.0 | 12 | 16.5 | 13.0 | 1.3 | 50.5 |
| N_total (%) | 77 | 0.24 | 0.15 | 0.07 | 1.20 | 22 | 0.22 | 0.10 | 0.10 | 0.45 | 12 | 0.22 | 0.06 | 0.13 | 0.29 |
| C_total (%) | 77 | 3.3 | 3.0 | 1.1 | 25.7 | 22 | 3.2 | 1.6 | 1.2 | 8.0 | 12 | 2.6 | 0.7 | 1.4 | 3.3 |
| C/N ratio | 76 | 13.3 | 2.2 | 9.8 | 19.9 | 22 | 14.0 | 2.6 | 11.3 | 19.9 | 12 | 11.8 | 0.6 | 10.7 | 12.7 |
| SOM (%) | 73 | 5.8 | 2.5 | 2.1 | 16.1 | 22 | 6.3 | 3.3 | 2.3 | 16.1 | 12 | 5.2 | 1.4 | 2.8 | 6.7 |
| P (mg kg$^{-1}$) | 76 | 280 | 122 | 93 | 693 | 22 | 234 | 70 | 111 | 353 | 12 | 321 | 119 | 204 | 613 |
| Ca (cmol kg$^{-1}$) | 21 | 11.4 | 8.0 | 2.6 | 37.1 | 5 | 6.9 | 4.1 | 2.6 | 11.4 | 4 | 11.7 | 5.9 | 7.7 | 20.4 |
| Mg (cmol kg$^{-1}$) | 21 | 2.3 | 1.4 | 0.5 | 6.1 | 5 | 1.5 | 0.8 | 0.5 | 2.7 | 4 | 2.6 | 1.1 | 1.8 | 4.1 |
| K (cmol kg$^{-1}$) | 21 | 0.45 | 0.26 | 0.09 | 1.29 | 5 | 0.34 | 0.21 | 0.09 | 0.60 | 4 | 0.46 | 0.27 | 0.29 | 0.85 |
| Na (cmol kg$^{-1}$) | 21 | 0.08 | 0.05 | 0.0 | 0.3 | 5 | 0.1 | 0.06 | 0.05 | 0.20 | 4 | 0.06 | 0.02 | 0.04 | 0.08 |
| Zn (mg kg$^{-1}$) | 21 | 2.4 | 3.0 | 0.1 | 9.7 | 5 | 0.5 | 0.4 | 0.2 | 1.2 | 4 | 4.1 | 4.2 | 0.6 | 9.7 |
| Cu (mg kg$^{-1}$) | 21 | 15.5 | 42.7 | 0.8 | 161 | 5 | 2.0 | 1.5 | 1.0 | 4.5 | 4 | 71.8 | 83.4 | 0.8 | 161 |
| Fe (mg kg$^{-1}$) | 21 | 117 | 62 | 10 | 232 | 5 | 158 | 73 | 69 | 232 | 4 | 91 | 38 | 63 | 143 |
| Mn (mg kg$^{-1}$) | 21 | 23.3 | 10.4 | 7.1 | 43.6 | 5 | 29.8 | 9.8 | 17.5 | 43.6 | 4 | 15.3 | 3.7 | 9.9 | 18.1 |
| pH | 77 | 5.0 | 0.7 | 3.8 | 6.6 | 22 | 4.9 | 0.8 | 3.8 | 6.4 | 12 | 5.2 | 0.3 | 4.9 | 5.7 |
| As (mg kg$^{-1}$) | 76 | 3.2 | 1.4 | 1.0 | 12.2 | 22 | 3.0 | 1.0 | 1.0 | 4.4 | 12 | 3.4 | 1.2 | 2.1 | 5.5 |
| Cd (mg kg$^{-1}$) | 76 | 0.39 | 0.13 | 0.18 | 0.74 | 22 | 0.37 | 0.12 | 0.18 | 0.60 | 12 | 0.43 | 0.13 | 0.27 | 0.66 |
| Cr (mg kg$^{-1}$) | 77 | 9.3 | 6.2 | 2.2 | 28.3 | 22 | 9.5 | 7.3 | 2.7 | 26.4 | 12 | 8.2 | 1.7 | 6.0 | 11.3 |
| Pb (mg kg$^{-1}$) | 77 | 5.6 | 2.1 | 2.8 | 10.7 | 22 | 5.6 | 2.1 | 2.8 | 8.5 | 12 | 5.3 | 1.2 | 3.9 | 7.3 |
| CEC (cmol kg$^{-1}$) | 21 | 14.2 | 9.3 | 3.4 | 44.6 | 5 | 8.8 | 4.6 | 3.4 | 13.6 | 4 | 14.8 | 7.0 | 9.9 | 25.0 |
| K$^+$/(Ca$^{2+}$+Mg$^{2+}$) | 21 | 0.04 | 0.02 | 0.02 | 0.1 | 5 | 0.04 | 0.03 | 0.02 | 0.07 | 4 | 0.04 | 0.03 | 0.02 | 0.09 |
| K$^+$/(Ca$^{2+}$+Mg$^{2+}$+Na$^+$) | 21 | 0.04 | 0.02 | 0.02 | 0.08 | 5 | 0.04 | 0.02 | 0.02 | 0.07 | 4 | 0.04 | 0.03 | 0.02 | 0.08 |
