# Peer review of "Back to the future? Conservative grassland management can"

_EGUsphere, 2022_

## Author Response (AR2)

**Replies to reviewers including listed changes**

**Reviewer 1**

Ina Säumel et al.

Author comment on "Back to the future- Conservative grassland management for Anthropocene soils in the changed landscapes of Uruguay?" by Ina Säumel et al., EGUsphere, https://doi.org/10.5194/egusphere-2022-335-AC1, 2022

**Response Reviewer 1 (*Please find our replies in italic*)**

Reply: Thank you for your helpful and constructive comments on our manuscript "Back to the future- Conservative grassland management for Anthropocene soils in the changed landscapes of Uruguay?". We really appreciate the time and efforts you did with the review. We respond your comments and suggestions point by point and explain how we addressed the issues in the manuscript.

This paper can make a beneficial contribution to literature. And it is well written. However, some improvements became apparently necessary.

*Reply: We are happy about this general valuation of our work and grateful for your suggestions to improve the manuscript.*

The use of the word "level" throughout the manuscript is often vague and unneeded. For instance, in L245, it would more advantageous for the reader to use "concentrations" instead of "levels". Similar when referring to acidity, one can see how omitting "level" actually enhances readability. Likewise, the frequent use of the word "value" is in several cases unnecessary and make the text wordy. Please revise the paper for conciseness where it makes sense based on these suggestions.

*Reply: We agree and revised the manuscript regarding the use of "level", "acidity", "value". We deleted "level" except in line 224, where we refer to the subcategory of level land use. We deleted "value" except in three sentences, if we refer to other reported background or reference data: in line 314 "The C/N ratio in topsoils ranged within values reported for grassland and timber plantations of the region"; in line 342, "our data on topsoil pH fall short compared to the values estimated by the FAO"; in line 358: "Some samples, especially from orchards and crops, exceeded the background values of copper, cadmium and arsenic "*

As noted the introduction is well written with a good style overall for readability; however, one still wonders towards the end of the intro section what the specific subject of the study is. What the fertility proxies are (?) – examples?

*Reply: Thank you for this comment. We add examples as suggested. In line 84: "we analysed i) the variation of fertility proxies such as soil organic carbon or content of nutrients,"*

L101 use "laboratory" as will be better than using "lab".

*Reply: Changed to laboratory.*

113 the assumptions that SOM= Corg x 2 can be directly challenged. A suggestion for the authors is to just report and discuss soil organic carbon – the statistical comparisons are actually the same. Organic C is the actual measurement in the study and statements made based on this metric will be more conclusive, concrete and hence insightful.

*Reply: We agree and changed this in the whole manuscript as follows: line 113: we deleted "and soil organic matter (SOM) as $C_{org}$ x 2 (Chenu et al. 2015)" and the reference in the reference list; line 194, 226, 232, 278, Figure 2,4,5 and the respective tables: SOM replaced by soil organic carbon (SOC).*

Check parentheses at L120

*Reply: changed to 0.01M. Thank you for careful reading.*

442 "livestock" spelling here and elsewhere (e.g., 442).

*Reply: corrected, thank you for careful reading.*

L170 How the classes of 'partially grazed' plots versus 'highly grazed' plots were established and applied to the study locations? – as part of the methodology.

*Reply: Thank you for this comment, we added this information in line 176ff. in the methodology chapter as follows: "We subdivided grassland plots according to the intensity of use: (i) undisturbed grassland (without grazing), (ii) partially grazed grasslands (with sporadic grazing and low animal charge), and (iii) highly grazed grassland (with high animal charge)."*

254 "farmers" watch the spelling.

*Reply: corrected. Thank you for careful reading.*

L266 The fact that riparian soils had higher concentrations of chemical properties than soils in other land uses does not equate to stating that organic carbon, cations and anions are moving from other land use to riparian soils; it just mean that they are higher. Tracing nutrients and carbon from certain land segments to other would require a different kind of study. Authors could state this as a new hypothesis; however, the evidence is not sufficiently compelling to indicate "Our data demonstrate the accumulation...". Likewise, based on this aspect, it becomes inadequate to refer to "trajectory" or "translocation" (e.g., L17)

*Reply: You are right, we do not proof it. It is a hypothesis. Changed as follows: Line 283ff. "Our data demonstrate the high amounts of organic carbon, nutrients and trace metals in topsoil samples from riverine forests, suggesting transport of soil particles from the surrounding land uses (e.g., grasslands, crop or timber plantations) to the borders of rivers, streams and creeks. Therefore, we assume that organic carbon, nutrients and trace metals are displaced within the landscape and accumulate in the floodplains. Regional soil erosion models estimate the loss of 2-5 tons ha-1 year-1 for a third of the country depending on precipitation, topography, soil erodibility and land management (Carrasco- Letelier and Beretta-Blanco 2017). One possible direct impact is the increasing eutrophication reported for larger local rivers, although the models used by these authors did not link Chlorophyll-a concentrations with agricultural land use (Beretta-Blanco and Carrasco-Letelier 2021 and replies)." and in Line 17: "We observed a loss of nutrients, trace metals and organic matter from grassland, crops and timber plantations, and assume that they accumulation in the topsoils of riverine forests, where high levels of nutrients, trace metals and organic matter are found." And in line 463: "Our soil survey data shows strong soil degradation of Uruguayan black soils from erosion, acidification and contamination, and suggests a translocation of nutrients and organic matter across the landscape from grassland, timber and crop plantations to the riverine forests."*

280 remove "the" before "half"

*Reply: Done.*

It becomes useful to acknowledge that inferences are being made based on a 0-to-10 cm depth increment. Other differences across land uses can be hidden deeper in the soil profile, and this is unknown based on this study.

*Reply: Thank you for this comment. We added in line 294: "However, other vertical processes and differences across land uses can be hidden deeper in the soil profile, and have not been analysed in this study."*

Authors can consider point out to fact that novel land uses (such as perennial grain cropping) may help to turn around land degradation into beneficial land aggradation. Kim et al. showed that these perennial grain cropping provide the advantage of perennial vegetation (somewhat resembling grasslands) of conserving and even enhancing short term and long term soil carbon storage and other ecosystems services. This study can also support that notion of soil degradation when chemical fertilizers are used (N fertilizer additions actually diminished soil C sequestration). This poses a change in land use systems across the regional landscape. In other words, instead of remaining locked in the existing land uses, one can look outside the box and find novel options.

Kim, K., Daly, EJ, M. Gorzelak, Hernandez-Ramirez, G. 2022. Soil organic matter pools response to perennial grain cropping and nitrogen fertilizer. Soil and Tillage Research. 220, 105376 https://doi.org/10.1016/j.still.2022.105376

*Reply: Thank you for this very valuable comment and the suggested reference to this interesting paper which was published recently. We added the following in Line 465: "Recent studies indicated that novel techniques such as perennial grain cropping can help to turn around cropland degradation into beneficial cropland aggradation by using the advantage of perennial vegetation of conserving and even enhancing short term and long- term soil carbon storage and other ecosystems services (Kim et al. 2022)."*

**Table 1.** It would be good to define that SD is standard deviation as each table needs to be standalone for interpretation.

*Reply: Added*

IT will be beneficial for the paper to use the expression "gravimetric moisture content" instead of "Humidity" in the Tables, method section 2.2 as well as elsewhere in the paper.

*Reply: Changed in line 106: "For gravimetric determination of moisture content..." and in the Tables we added an explanation in the Table header as the long name would crash the tables. We added: 'humidity' stands for the gravimetric moisture content. End.-*

**Reviewer 2**

Ina Säumel et al.

Author comment on "Back to the future- Conservative grassland management for Anthropocene soils in the changed landscapes of Uruguay?" by Ina Säumel et al., EGUsphere, https://doi.org/10.5194/egusphere-2022-335-AC2, 2022

**Response Reviewer 2 (*Please find our replies in italic*)**

A one-time survey across anthropogenic soils in Uruguay was carried out and analyzed for different soil parameters with the aim to identify interaction between land use and

soil characteristics. The manuscript covers the relevant and timely issue of non-sustainable land use practices that support soil loss and degradation.

*Reply: We are happy that you see the relevance of the topic of our manuscript. Thank you for your time and willingness to review our manuscript.*

However, I am missing a substantial contribution to scientific progress in the methodology and/or results. There are long-term and recurring topsoil surveys established worldwide, results are being published in report formats on a regular basis, and in general, these data sets (especially when run over long time periods) hold valuable information for various scientific questions. However, in this manuscript, the authors did not communicate the aim, research question or hypothesis addressed with the study. A purely exploratory statistical analysis of a set of standard analyses of soil samples is not sufficiently novel or unique for a publication in SOIL. Therefore, I regret I cannot recommend the publication of this manuscript.

*Reply: We regret that you have the impression that our manuscript does not make a substantial contribution to scientific progress. Of course, we as authors and conductors of the research and the funders of our study have a different perspective. However, we take your critics seriously and highlight now better the novelty and contribution of our manuscript.*

*Long-term and recurring topsoil surveys that covers such a great variety of different land uses (different types of native forests, grassland, timber plantations, crops) are rare in the global south and especially in countries with limited resources. We need these surveys especially in those countries were land use change and overexploitation for the global market takes place right now. In Uruguay, these changes happen in remote areas with voiceless people and with governments that have no resources to control or reclaim possible impacts. Independent data are crucial to discuss the local agendas and discourses of different stakeholders. We have already pushed the finger on this point please see line 62-71: "As soil degradation is extremely relevant for countries like Uruguay, which are socioeconomically dependent on their soils (Zubriggen et al. 2020), it is a topic of discussion for local farmers, academia, and the public. An actualization of the state of the art of soils and related processes is needed (García-Préchac et al. 2004; De Faccio et al. 2021), particularly as there has been little study of the impacts of the Uruguayan grassland intensification on soils properties (Beretta-Blanco et al., 2019). At the same time, while a new paradigm for grassland intensification with a wide set of means including fertilization has been proposed to increase economic and environmental sustainability (Jaurena et al., 2021), it is urgent to get more insights into the dynamics of nutrient in soils of Uruguay and their availability for crops (Beretta-Blanco et al., 2019)." And in line 77f. "We contribute to a better understanding of globally occurring degradation processes in the field of tension between desired soil productivity, yield limits, especially in erosion sensitive soils, and necessary soil conservation."*

*We reworked the part of the objectives of our study in the introduction as follows (line 81-91): "We therefore explored soil parameters describing current chemical conditions of topsoils that are parts of different soil groups and orders, and Uruguayan soil categories. Specifically, in order to explore the gains and losses of macro and micro-nutrients and soil organic carbon across landscapes and to determine the impact of land use change on acidification and trace metal mobility and related trade-offs with soil degradation and conservation. In detail we address the following question: i) how do fertility proxies such as soil organic carbon and content of nutrients, acidification (pH) and trace metals accumulation in topsoils vary across different land uses? Thus, we expand the knowledge across land uses from more natural to strongly modified uses and discuss the results in light of different degradation processes such as erosion, depletion of nutrients or carbon, acidification and accumulation of pollutants and in the light on current debates on intensification."*

*We also checked again if there have been publications on surveys that we might have overseen unintentionally before. Topsoil surveys that cover such a great variety of different land uses are missing. We explained this situation in line 62-71. "As soil degradation is extremely relevant for countries like Uruguay, which are socioeconomically dependent on their soils (Zubriggen et al. 2020), it is a topic of discussion for local farmers, academia, and the public. An actualization of the state of the art of soils and related processes is needed (García-Préchac et al. 2004; De Faccio et al. 2021), particularly as there has been little study of the impacts of the Uruguayan grassland intensification on soils properties (Beretta-Blanco et al., 2019). At the same time, while a new paradigm for grassland intensification with a wide set of means including fertilization has been proposed to increase economic and environmental sustainability (Jaurena et al., 2021), it is urgent to get more insights into the dynamics of nutrient in soils of Uruguay and their availability for crops (Beretta-Blanco et al., 2019)." Regarding existing surveys and the novelty of our data we already explained in line 310-317: "This is clearly reflected in the results of our topsoil survey, which also adds interesting insights from timber plantations, grasslands and native forests to an existing database consisting mainly of crops and pastures samples from 2002-2014, which demonstrated the loss of organic matter by 25% and an increasing loss of nutrients (Beretta-Blanco et al. 2019). We contribute deeper insights on fertility, acidification and trace metals accumulation in topsoils from a wide range of different land uses, which is, to our knowledge, unique for the region since the CONEAT classification (CONEAT Index, 1976)." Or in line 434ff: "To our knowledge, there has been no regional study of trace metals in the native riverine forests or timber plantations. Our work thus expands the evidence base for these land uses."*

*Concerning your second point of long-time data, we totally agree with you, that long term data are very valuable and necessary, especially those that follows a standardized protocol. We call this always as the "gold of ecology". Regarding this we only want to state that the reality of funding and research projects looks different, research projects are often only funded for 2-3 years, in this time you can maybe establish a sampling design and do one or maximum two sampling runs and you have to publish the results immediately in high score journals to get in a new call funding for next projects. In addition, we had two years of funding without any possibility to go back to the field for sampling.*

Some points that I would like to share to reason my decision, and that might be helpful for future manuscript preparations, follow below.

*Reply: Thank you for your suggestions to improve our manuscript.*

I stumbled a bit over the title, being a question "sentence" without a verb. Titles, especially when they include questions, should be concise and meaningful.

*Reply: Thank you for this comment. Changed as follows: Back to the future? Conservative grassland management can preserve soil health in the changing landscapes of Uruguay*

It is not clear in the abstract, what the goal of the study was (to "understand" soils), and what was done ("we analyzed ..."). From reading the abstract, I wondered whether this study is about a spatial and/or time series analysis. The results mentioned in the abstract give clues to sophisticated analysis methods: loss in nutrients, accumulation in topsoils on riverine forests, translocation, local land use trajectories. But later in the manuscript it appears that none of this was actually investigated directly.

*Reply: We see recognize that are abstract needs improvement and we changed the abstract to meet the comments also from reviewer 1: We analysed agri- and silvicultural intensification of Uruguayan grasslands in a country wide survey on fertility proxies, pH*

and trace metals in topsoils originating from different land uses. We are convinced that we did a sophisticated analysis of the data we have. We included the important points based on the discussion. Our discussion is based on our data and on evidence in literature.

Changed as follows: The 'soils of the anthropocene' are predominately agricultural. To understand them, we analysed agri- and silvicultural intensification of Uruguayan grasslands in a country wide survey on fertility proxies, pH and trace metals in topsoils originating from different land uses. We observed a loss of nutrients, trace metals and organic matter from grassland, crops and timber plantations. As an example, the cation exchange capacity was 160 percent higher in native forests compared to grasslands and lowest in timber plantations, reaching only half of the CEC in grasslands Acidification of topsoils continues as three fourth of all samples are 'extremely acidic' and 'very strongly acidic' and lowest in timber plantations. Topsoils of riverine forests accumulate more trace metals compared to the other uses. We assume an accumulation in the topsoils of riverine forests, where high levels of nutrients, trace metals and organic matter are found. The translocation of nutrients and organic matter across the landscape to the erosion base depends on local land use trajectories. Increasing soil acidification is driving a positive feedback loop, and land use intensification is leading to degradation of local black soils within a few decades. Our data raises questions about the resilience and carrying capacity of Uruguayan soils with regard to currently implemented highly productive management forms, including the use of timber plantation for carbon sequestration, and supports more conservative forms of extensive management on the grassland biome.

This is also changed in the discussion section 4.1 as follows: Our data demonstrate the high amounts of organic carbon, nutrients and trace metals in topsoil samples from riverine forests, suggesting transport of soil particles from the surrounding land uses (e.g., grasslands, crop or timber plantations) to the borders of rivers, streams and creeks. Therefore, we assume that organic carbon, nutrients and trace metals are displaced within the landscape and accumulate in the floodplains. Regional soil erosion models estimate the loss of 2-5 tons ha-1 year-1 for a third of the country depending on precipitation, topography, soil erodibility and land management (Carrasco-Letelier and Beretta-Blanco 2017). One possible direct impact is the increasing eutrophication reported for larger local rivers, although the models used by these authors did not link Chlorophyll-a concentrations with agricultural land use (Beretta-Blanco and Carrasco-Letelier 2021 and replies). However, other vertical processes and differences across land uses can be hidden deeper in the soil profile, and have not been analysed in this study.

Regarding the trajectories of land use change we included more detailed information in the section Study area and design as follows: Line 129-143: "If the owner agreed, plot selection was stratified by different rural land use types: grassland, timber plantations of Pinus and Eucalyptus species, native forest, and crops. Native forests cover mainly riverine and park forests. The later are a savanna like transition zones between riverine forests and the open grasslands. We subdivided grassland plots according to the intensity of use: (i) undisturbed grassland (without grazing), (ii) partially grazed grasslands (with sporadic grazing and low animal charge), and (iii) highly grazed grassland (with high animal charge). Land use change from 1986 to 2017 follows basically three different trajectories: i) the expansion of timber plantations over grassland leading to a disaggregation of grassland by timber plantations; ii) cropland expansion where crop cover maintains the open landscape character of former grasslands, grassland conservation where large and regularly interconnected riverine forests in a landscape dominated by grasslands (Ramírez and Saümel 2021) and grassland intensification changing from natural grassland to so called 'improved' or artificial grasslands (Modernel et al. 2016; Jaurena et al. 2021). Fertilization and application of other agrochemicals is standard procedure in timber plantations, artificial grasslands and industrial crops."

Some parts in the manuscript are rather misleading: In the methods it is mentioned that landowners were asked permission for a long-term monitoring of their land, but there is no further information found on this in the remaining text.

*Reply: Here we do not understand the point of the reviewer comment. We mentioned that asking the owner to get permission was import to install monitoring sites and take samples and take again samples in the future to get more data as 'gold of ecology' as discussed above. This shaped the location of the plots and sampling sites, thus we mentioned it in M&M, but there was no need to mention this again.*

The discussion actually starts with setting the background for a rationale to carry out the survey, so it is somewhat out of place here and should be part of the introduction.

*Reply: As you suggested we highlight the need for our contribution now more pronounced in the introduction, please see also our comments and changes above.*

Some parts are misleading or hard to understand: e.g., the authors discuss the role of riverine forests, but results are shown for native forests. Sometimes, "park forests" are also mentioned without an explanation of that term.

*Reply: Thank you for these comments. We see this point and provide now more details on local ecosystems, land uses. In Uruguay the dominating native forest type are riverine forest, there are also in some parts hill forests. The park forests are a savana like transition form from riverine forest to grassland. The local discussion to which extent these park forest are natural or culturally shaped (e.g. as a result of grazing) is ongoing since decades without agreement. We added more information and references on the land use types in the methodology part as follows: (see Line 129-143: "If the owner agreed, plot selection was stratified by different rural land use types: grassland, timber plantations of Pinus and Eucalyptus species, native forest, and crops. Native forests cover mainly riverine and park forests. The later are a savanna like transition zones between riverine forests and the open grasslands. We subdivided grassland plots according to the intensity of use: (i) undisturbed grassland (without grazing), (ii) partially grazed grasslands (with sporadic grazing and low animal charge), and (iii) highly grazed grassland (with high animal charge). Land use change from 1986 to 2017 follows basically three different trajectories: i) the expansion of timber plantations over grassland leading to a disaggregation of grassland by timber plantations; ii) cropland expansion where crop cover maintains the open landscape character of former grasslands, grassland conservation where large and regularly interconnected riverine forests in a landscape dominated by grasslands (Ramírez and Säumel 2021) and grassland intensification changing from natural grassland to so called 'improved' or artificial grasslands (Modernel et al. 2016; Jaurena et al. 2021). Fertilization and application of other agrochemicals is standard procedure in timber plantations, artificial grasslands and industrial crops.").*

The CONEAT index or categories are nowhere explained, the mere numbers are meaningless to the reader not familiar with this classification.

*Reply: We provide now more details in the method section as follows: line 197ff.: "The CONEAT groups are defined by their productive capacity in terms of beef, sheep and wool expressed by an index relative to the average productive capacity of the country, to which the index 100 corresponds. The classification is based on photo-interpretation at a scale of 1:40,000, field verifications and physico-chemical analysis of the soils. The productivity indices correspond to soil groups. The CONEAT groups have been defined by the dominant and associated soils according to the Soil Classification of Uruguay. The groups are related to the units of the Soil Reconnaissance Chart of Uruguay at a scale of 1:1,000,000. For each group, some important soil properties and associated landscape characteristics are indicated. The nomenclature of the CONEAT groups correlates with the*

*Soil Use and Management Zones of Uruguay. The Soil Groups are superimposed on the rural parcel and are represented in the CONEAT cartography at a scale of 1:20,000 (for more details see MGAP, 2020)."*

The discussion is mostly very vague and I cannot find new insights or sound conclusions

from the performed analysis results.

*Reply: We would appreciate very much if you come up with more concreate examples. We did our best to be as precise as possible and within our data and other evidence reported in literature. We changed parts of the discussion following the suggestions of reviewer 1.*

Finally, the statement of the last paragraph, that extensive management of native grassland would be better for soil fertility/health is nothing that is new or concluded from the presented study results. A final conclusion is missing.

*Reply: The paragraph 4.5 is a positioning within the debate on the future of grassland management based on the insights from our data. Grasslands are globally seen as cropland reserve e.g. to fight hunger (SDG2). Extensively used grasslands are often seen as unproductive and intensification strategies are discussed. Our data do not support intensification strategies.*

*Thank you for your suggestion and added now a new paragraph on conclusions as follows: "The land use intensification in Uruguay associated with increasing inputs of energy, nutrients and pesticides leads to an overall loss of soil fertility and increasing toxicity related to acidification, salinization and trace metal contaminants. Our data demonstrate the high amounts of organic carbon, nutrients and trace metals in topsoil samples from riverine forests, suggesting transport of soil particles from the surrounding grasslands, crop or timber plantations to the borders of rivers, streams and creeks. Of all the fertility proxies assessed, phosphorus in topsoil was most significantly affected by different land uses, being highest in native forests. Cation exchange capacity was also highest in native forests and lowest in timber plantations, where only half that of grasslands was measured. Our study highlights that soil acidification is ongoing and probably also mobilizing trace metals and their accumulation in riverine forest topsoils."*

---

## Author Response (AR3)

SOIL R2

Overall these authors have presented a wide-ranging dataset on Uruguayan soils as they relate to both soil type and land-use type. I commend the authors for this large-scale effort and attempt to present this large and complex dataset. They have drawn some interesting conclusions and comparisons such as the fact that authors posit the movement of surface soils from surrounding grasslands into riverine soils. There is, however, a general lack of clarity. As a reader it is not clearly presented what is being compared, and what are results of native differences to soils due to soil forming factors and ecology, and what is due to anthropogenic influence. For example, forests are discussed, but are any of the forests managed intensively? I think the authors could refocus the writing as well as the data presented on the most relevant comparisons in their opinions. I do believe this is worthy of publication in SOIL, but there are significant issues that need to be addressed, see below.

*Reply: Thank you for your time dedicated to review our manuscript. Based on your helpful suggestions we revised the manuscript and hope we presented in the revised version in a better way to enhance clarity what is being compared, what is related to soil forming factors and ecology, and what is due to anthropogenic influence. Timber plantations are managed intensively. Native forests are protected by law in Uruguay. In fact, we would prefer to use the term "forests" mainly for the native forests, which are mostly riverine or gallery forests or to a lesser extent hill and park forests. The later are a transition between the riverine forests and the open savannas. In contrast the Eucalyptus plantations with rotation cycles of 7 to 10 years cannot be compared to planted timber forests in the north that have life cycles over decades and establish as forest like ecosystems. Moreover, our study is limited to the topsoils, whereas soil classification focuses on the whole soil profiles. In regard to soil formation factors, we also checked for spatial autocorrelation (Table S2) and we include the analysis of the different soil classes in Uruguay (e.g. 2.3 and at the beginning of the result section and in the section 3.2). We added the following in the abstract: "The 'soils of the anthropocene' are predominately agricultural. To understand them, we analysed agri- and silvicultural intensification of Uruguayan grasslands in a country wide survey on fertility proxies, pH and trace metals in topsoils originating from different land uses across the whole country. Thus, our results reflect interactions of both the natural diversity of the Uruguayan soil formation and impacts of land use change." The introduction starts with: "Human activities alter the bio- and pedosphere, leaving a footprint of such a magnitude that it can be verified stratigraphically (Waters et al., 2016). This unprecedented transformational force is intimately related to the expansion of societies and its productive frontiers, causing a loss of biodiversity, habitat and soil degradation and, consequently, to ecosystem modification (Foley et al., 2005, Borrelli et al., 2017). In this context, soil sciences have transitioned from studies on natural soil formation to the science of 'anthropedogenesis' (Richter, 2020), focussing on the 'soils of the anthropocene' that are predominately agricultural (51 Million km2) or urban (1.5 Million km2; FAO, 2019)." In addition, we added in the last paragraph of the introduction: "In detail we address the following question: how do fertility proxies such as soil organic carbon and content of nutrients, acidification (pH) and trace metals accumulation in topsoils vary across different land uses (i.e. comparing grassland, timber plantation, native forest, and agricultural land)?"".*

Some of the major differences between treatments that authors point out are between forests and grasslands, but the wording of the article makes it seem like the results observed come from human-induced land use changes. Are there not native grasslands in Uruguay that have inherently different characteristics than the forests? It has been observed in many ecosystems that forest soils are the most "healthy" CHECK THIS, so highlighting differences between two natural landscapes (grasslands and forests) doesn't seem necessary here or at least should not be a focus, it should be the difference between soils under heavy anthropogenic influence that should be the

focus (timber plantations, agricultural crops).

*Reply: Thank you for this comment, actually the grasslands in Uruguay have been always used and impacted by humans and purely natural grasslands are very scarce. As an example, today within the fenced timber plantations we have grassland plots that are currently without grazing, but have a grazing history in the past. We explain the situation in the description of the study site chapter 2.1 as follows: "Land use change from 1986 to 2017 follows basically three different trajectories: i) the expansion of timber plantations over grassland leading to a disaggregation of grassland by timber plantations; ii) cropland expansion where crop cover maintains the open landscape character of former grasslands, grassland conservation where large and regularly interconnected riverine forests in a landscape dominated by grasslands (Ramírez and Säumel 2021) and grassland intensification changing from natural grassland to so called 'improved' or artificial grasslands (Modernel et al. 2016; Jaurena et al. 2021)."*

Abstract:
L17-20 " As an example, the cation exchange capacity was 160 percent higher in
18 native forests compared to grasslands and lowest in timber plantations, reaching only half of
19 the CEC in grasslands acidification of topsoils continues as three fourth of all samples are
20 'extremely acidic' and 'very strongly acidic' and lowest in timber plantations. "

Not clear what you mean by "lowest in timber plantations" do you mean the lowest pH which is more acidic or do you mean the lowest "level of acidity"

*Reply: Thank you for this comment, there was a mistake possibly due to a copy paste error in the final document. We are very sorry for this. In order to enhance clarity, we changed this as follows: "As an example, the cation exchange capacity was 160 percent higher in native forests compared to grasslands and lowest in timber plantations, reaching only half of the CEC in grasslands. Acidification of topsoils continues as three fourth of all samples are 'extremely acidic' and 'very strongly acidic'."*

L38-39 "Mio." what does this mean?
*Reply: we do not use the abbreviation to avoid misunderstanding: Changed to Million.*

L73 replace "nutrient" with "nutrients"
*Reply: changed accordingly, thank you for your careful revision.*

L75-78 run-on sentence and not completely clear to reader, please split it up/reword it.
*Reply: We suppose that the comment is related to this sentence, which is now splitted into two: "Soil classifications are mainly based on subsoils. However, we focus on topsoil as the most relevant and very responsive interface for ecological processes and farmer's management. Understanding the state of the art of topsoils and its processes is crucial for developing recommendations for sustainable land management practices."*

L80-83 "We contribute to a
81 better understanding of globally occurring degradation processes in the field of tension between
82 desired soil productivity, yield limits, especially in erosion sensitive soils, and necessary soil
83 conservation."
This sentence does not make sense to me, please explain what you mean by "field of tension"
*Reply: Thank you for this comment. We changed this as follows: "We contribute to a better understanding of globally occurring degradation processes among often conflicting goals such as desired soil productivity, yield limits, especially in erosion sensitive soils, and necessary soil conservation."*

L88 not clear how trace metal mobility was measured in this study, trace metal presence was clear

from the methods and results, but not their mobility, or do you mean cation exchange capacity in general?

*Reply: Thank you for this comment. We changed this as follows: "Specifically, in order to explore the gains and losses of macro and micro-nutrients and soil organic carbon across landscapes and to determine the impact of land use change on acidification and trace metal presence and related trade-offs with soil degradation and conservation."*

L89-94 you begin to make a list with the numeral "(i)" but then make the list as a sentence with commas, either put numerals next to each measured parameter, or leave the numerals out and present the sentence as a list of measured parameters

*Reply: Thank you for this comment. We changed this as suggested (without numerals).*

L89-94 it would also be useful for the reader to restate the major comparisons made in the study (the varying combinations between grassland, forest plantation, forest, and cropland)

*Reply: Changed as suggested: "In detail we address the following question: how do fertility proxies such as soil organic carbon and content of nutrients, acidification (pH) and trace metals accumulation in topsoils vary across different land uses (i.e. comparing grassland, timber plantation, native forest, and agricultural land)?"*

Methods:

L 102-105 please specify how many plots and samples of each land use type was collected, assuming they are equally distributed 280 samples/4 treatments = 70 samples in each treatment? So n = 70? And how many plots of each treatment? That is not clear from this sentence. Please also specify the numbers of samples collected in each sub-category. Since there are many different land uses and sub categories, I would suggest putting this information into a table that specifies land use, land use sub category, a brief explanation of that land use, number of plots and samples within each category and sub category, and total number of samples for each category and subcategory that are analyzed statistically. There are many different treatments and sub treatments and as they are written it is hard to understand how many of each were sampled and what was analyzed in each one. While this information is present within Table 1 and Tables A1-A4, it is hard to pick out. Would it be possible to say how many samples were collected in each treatment in a simple table?

*Reply: Yes, the sample numbers per treatments are provided in the tables of the appendices, we added this information into Table 1. Unfortunately, we have no equal distribution of samples per land use types as a second shipment did not get the allowance for export and importation. The N for subtypes can be find in the supplements… Fig. S3*

L121-123 you state here that the edges of plots were sampled, why not sample in the middle of the field? Edges are typically avoided in soil sampling in order to get a representative sample of a plot, how far from the edge of the plot was sampled?

*Reply: The rectangular plots are placed in homogeneous areas of each land use to avoid edge effects. Thus, our sampling was at the virtual edges of our plots but without edge effects. We added the sentence: ". The plots are placed in homogenous areas to avoid edge effects."*

L126-127 how big were the pieces taken out? Ones that did not pass the 2mm sieve? Removing organic matter from soil can significantly affect the percentage of organic matter and the C/N ratio

*Reply: Yes. This is a normal procedure in soil labs. Soil samples were sieved through a 2-mm mesh screen to remove, roots, and debris prior to analysis.*

L128-129 which samples were analyzed for soluble cations and micronutrients?

*Reply: We also added this information in Table 1. A detail for the samples analyzed for soluble cations and micronutrients is described in APPENDIX A, in Tables A1 and A2.*

L133-135 which methods, specifically, were used to determine carbonate and SOC content?

***Reply: We added the methods in line 130f. as follows: "We analysed 280 samples regarding macronutrients, pH and trace metals and 80 samples for soluble cations and micronutrients (Table 1; Sadzawka et al. 2006; Zagal & Sadzawka 2007)."***

L136-139 include superscript charges for each ion
***Reply: done as suggested***

L182-186 were any outliers removed, or were they only identified? If any were removed you might add that justification based on experimental conditions in your supplemental material.
***Reply: This information can be found in Fig.S2 and added the main text line 190ff. as follows: "We tested for outliers using the 1.5-3 IQR threshold and the function outlierTest from the R package car (Fox and Weisberg 2019), reviewing the flagged observations case by case in the experimental context. The outliers were removed (Supplementary Material: Fig.S2)."***

L195 is there a reference for Benjamini and Hochberg?
***Reply: Added.***

Results
L214-222 - you rank the highest and lowest values in addition to the mean, but there are not statistical results presented in any of the referred tables (Table 1, A1-A2) here please include what is statistically significant.
***Reply: in this part we only describe the general ranges values. Significant differences between land use types are shown in Figure 4-5, 6 and described in the next sections.***

Figures:
General: make fonts larger, hard to read
1a - numbers inside of pie charts are hard to read - increase font size of pie chart numbers
***Reply: We re-arranged the Figure 1 to enhance readability***

1d - not clear what the CONEAT numbers are referring to, and what do the letters next to numbers mean in the legend? I see you describe these around L 167 but having a short description in this caption to help the reader remember would be helpful
***Reply: we added a short explanation here and refer to the main text as follows: "The Uruguayan CONEAT index provides a detailed classification that takes into account soil type, texture, natural vegetation, altitude and geology (see details in chapter 2.3)."***
***In the Chapter 2.3: this is explained in detail: "We intersected the coordinates of the centre of the plots with maps containing geospatial information on the classification of the Uruguayan soils using ArcGIS 10.3 (ESRI, 2018). For Soil Groups classification, we used the of the World Reference Base for Soil Resources (WRB; IUSS Working Group, 2015); for Soil Orders, the USDA soil taxonomy (Soil Survey Staff, 1999); and for the local Uruguayan classification, Soil CONEAT (Comisión Nacional de Estudio Agronómico de la Tierra) Groups categories, which include productive capacity of cattle and sheep (MGAP, 2020). The CONEAT groups are defined by their productive capacity in terms of beef, sheep and wool expressed by an index relative to the average productive capacity of the country, to which the index 100 corresponds. The classification is based on photo-interpretation at a scale of 1:40,000, field verifications and physico-chemical analysis of the soils. The productivity indices correspond to soil groups. The CONEAT groups have been defined by the dominant and associated soils according to the Soil Classification of Uruguay. The groups are related to the units of the Soil Reconnaissance Chart of Uruguay at a scale of 1:1,000,000. For each group, some important soil properties and associated landscape characteristics are indicated. The nomenclature of the CONEAT groups correlates with the Soil Use and Management Zones of Uruguay. The Soil Groups are superimposed on the rural parcel and***

*are represented in the CONEAT cartography at a scale of 1:20,000 (for more details see MGAP, 2020)."*

Figure 2:
The caption leads reader to believe that there are three subfigures (alb, c) but this is a single figure, can you clarify if there are supposed to be additional figures here? It appears the information you state is there, but this is a formatting issue.
***Reply: Sorry, we changed the Figure accordingly adding the missing (a), (b) and (c).***

"Colour intensity and the size of the circle are
741 proportional to the correlation coefficients (ρ)" The size and the color of circles is proportional to correlation coefficients? The smallest circles are all blue, but the orange circles are lower (read - negative) this is not worded clearly, please be specific about what the size and color represent. Perhaps you mean the size of the circle refers to the absolute value of the correlation coefficient and the color refers to the direction (positive or negative) of the correlation.

Also a comment on methods: Fig.3 what do the colored circles and oblong shapes represent? A certain statistical threshold?
***Reply: Thank you, we changed this accordingly.***